**RESEARCH**                                                                    **Open Access**

# Genomic basis underlying the metabolome-mediated drought adaptation of maize

Fei Zhang[1,2†], Jinfeng Wu[1,2†], Nir Sade[3], Si Wu[4], Aiman Egbaria[3], Alisdair R. Fernie[5], Jianbing Yan[1,2], Feng Qin[6], Wei Chen[1*], Yariv Brotman[5,7*] and Mingqiu Dai[1,2*]

* Correspondence: chenwei0609@ mail.hzau.edu.cn; Brotman@mpimp-golm.mpg.de; mingqiudai@mail. hzau.edu.cn
†Fei Zhang and Jinfeng Wu contributed equally to this work.
[1]National Key Laboratory of Crop Genetic Improvement, Huazhong Agricultural University, Wuhan 430070, China
[5]Max Planck Institute of Molecular Plant Physiology, 14476 Potsdam, Germany
Full list of author information is available at the end of the article

## Abstract

**Background:** Drought is a major environmental disaster that causes crop yield loss worldwide. Metabolites are involved in various environmental stress responses of plants. However, the genetic control of metabolomes underlying crop environmental stress adaptation remains elusive.

**Results:** Here, we perform non-targeted metabolic profiling of leaves for 385 maize natural inbred lines grown under well-watered as well as drought-stressed conditions. A total of 3890 metabolites are identified and 1035 of these are differentially produced between well-watered and drought-stressed conditions, representing effective indicators of maize drought response and tolerance. Genetic dissections reveal the associations between these metabolites and thousands of single-nucleotide polymorphisms (SNPs), which represented 3415 metabolite quantitative trait loci (mQTLs) and 2589 candidate genes. 78.6% of mQTLs (2684/ 3415) are novel drought-responsive QTLs. The regulatory variants that control the expression of the candidate genes are revealed by expression QTL (eQTL) analysis of the transcriptomes of leaves from 197 maize natural inbred lines. Integrated metabolic and transcriptomic assays identify dozens of environment-specific hub genes and their gene-metabolite regulatory networks. Comprehensive genetic and molecular studies reveal the roles and mechanisms of two hub genes, *Bx12* and *ZmGLK44*, in regulating maize metabolite biosynthesis and drought tolerance.

**Conclusion:** Our studies reveal the first population-level metabolomes in crop drought response and uncover the natural variations and genetic control of these metabolomes underlying crop drought adaptation, demonstrating that multi-omics is a powerful strategy to dissect the genetic mechanisms of crop complex traits.

**Keywords:** Maize, Drought tolerance, Metabolome, Natural variation, Stress responses

## Introduction

Drought is a major environmental stress that threatens crop survival and yields, with critical effects on human society. In human history, drought even caused the collapse of several civilizations [1]. It is predicted that the world population will reach ~ 9 billion in 2050, and by then, there will be 70% more food demands than today (http://www.fao.org/wsfs/world-summit/en). Maize (*Zea mays*) is a major cereal crop in the world due to its nutritional value and high yield. By 2050, maize is expected to supply more than 50% of the cereal demands to feed the increasing population. Although maize has been promoted in yield during the past decades, its drought susceptibility has also increased concomitantly [2]. Moreover, global warming and extreme climate change are increasing the frequency of drought in arable land [3]. Therefore, there is a central interest in study of how to improve crop drought tolerance through genetic and molecular approaches.

The plant drought-tolerance trait is complex and regulated by many quantitative trait loci (QTLs) with minor effects [4]. To understand the genetic basis of maize drought tolerance, a number of QTLs associated with plant biomass, height, and anthesis-silking interval (ASI) in the drought response have been identified by linkage or association mapping strategies [5, 6]. The survival rate after severe drought stress, which largely indicates the activation of water-saving mechanism in plant under drought stress, has been widely used as an index to clone genes in plant drought tolerance. Recently, several maize drought-tolerant genes have been cloned and functionally studied by candidate association analyses or genome-wide association studies (GWAS) with the index of survival rate [7–10]. So far, most studies about maize drought tolerance have been focused on a few easy-to-measure traits. However, drought causes numerous biochemical and physiological responses. Our knowledge of what these responses are and how they are genetically controlled is very limited.

Metabolites are small molecules and perceived the end products of metabolite processes and physiological pathways. The metabolite profiling based on gas chromatography (GC) or liquid chromatography (LC) coupled with mass spectrometry (MS) is a powerful approach to study the metabolic responses of plants to various environmental stresses [11]. Studies with metabolite profiling have shown that numerous structurally different metabolites are produced in plants in response to various environmental stresses, including cold, heat, and salt stresses [12–14]. Several important metabolites and their associated genes have been shown to play roles in plant drought tolerance [15–18]. Recently, the metabolic responses of maize plants to drought stress in several tissues have been revealed using inbred or hybrid lines [19–21]. Several QTLs were found to be associated with several metabolites in drought response [22]. In addition, most studies of crop population-level metabolomes, including metabolite profiling and genetic dissection, have been focusing on populations grown under normal condition. However, the natural variations and genetic control of population-level and genome-wide metabolites underlying plant environmental stress adaptation, including those in maize drought adaptation, remain elusive.

In this study, we performed genome-wide metabolite profiling of a diverse maize association panel consisting 385 natural inbred lines [23, 24]. We identified 3890 metabolites, 1035 of which were differentially accumulated in maize plants under well-watered (WW) and drought-stressed (DS) conditions, representing effective indicators of maize

drought response and tolerance. The genetic control of these drought-responsive metabolites was explored. We further verified two hub candidate genes, *Bx12* and *ZmGLK44*, for their role in regulating benzoxazinoids and tryptophan biosynthesis as well as maize drought tolerance via comprehensive genetic and molecular methods. Our results provide novel insights into the genetic bases of metabolome-mediated drought adaptation in a crop.

## Results

### A drought-responsive metabolome landscape of a global diverse maize association panel

To understand how metabolites function in maize drought response and tolerance, we grew a maize set of 385 natural inbred lines in a greenhouse under WW and DS conditions (Additional file 1, Figure S1a; see "Materials and methods") [24]. This maize set consisted of 135 temperate inbred lines, 153 tropic/subtropic lines, and 97 lines with mixed origin, showing broad, diverse genetic compositions (Additional file 2, Table S1) [24]. Leaf samples were harvested and subjected to liquid chromatography- / high-resolution mass spectrometry (LC-HRMS) analysis with MTBE extracts, which can measure a broad range of metabolites. After the metabolites detected in less than 80% samples were filtered out, a total of remaining 3890 nonredundant metabolites were finally extracted from 770 libraries of the 385 inbred lines grown under both WW and DS conditions, including 353 putatively annotated metabolites (Fig. 1a; Additional file 2, Table S2). We conducted principal component analysis (PCA) for the maize inbred lines with all these 3890 metabolites. The results showed a clear separation between the WW and DS conditions, with that principal component (PC) 1 explaining the

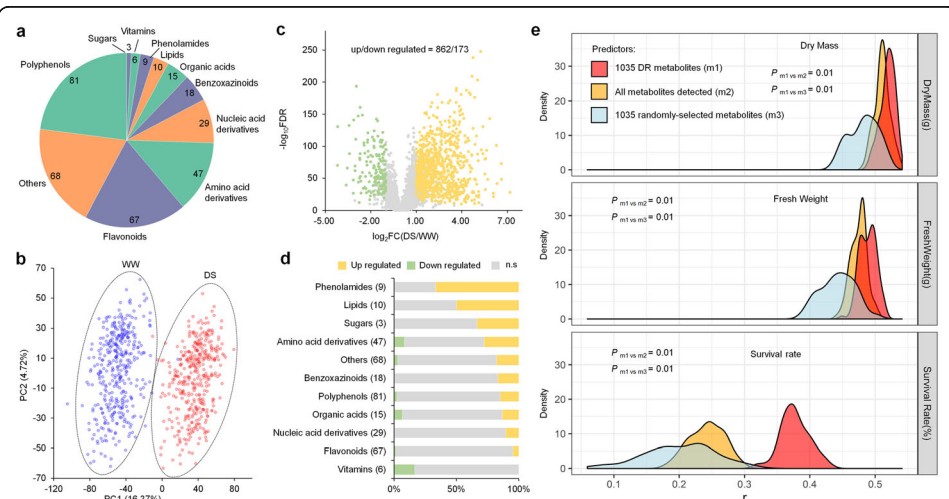

**Fig. 1** Identification of the metabolites. **a** Classes of the metabolites with annotated structures. **b** PCA of the 385 maize inbred lines based on the metabolites identified under WW and DS conditions. The inbred lines grown under WW and DS conditions are indicated as blue and red dots, respectively. **c** Volcano plot showing the downregulated (green dots) and upregulated (yellow dots) metabolites under DS conditions compared to those under WW conditions. **d** Proportions of the up- (yellow), downregulated (green), and non-affected (gray) metabolites with known structures. **e** Comparison of the prediction accuracies of dry masses, fresh weights and survival rates by all, drought-responsive, and randomly selected metabolites in the maize population after drought stress via the rrBLUP approach. Statistical significance was determined by a bootstrap significance approach (see "Materials and methods")

largest proportion (16.37%) of the observed phenotypic variance (Fig. 1b). These data indicated that leaf metabolites in the maize population were very sensitive to drought. In line with our results, a previous study showed that drought had a great effect on metabolite production in developing maize kernels [21].

To gain insights into the metabolic responses of maize to drought, we combined different assays to investigate the changes of all the detected 3890 metabolites in the population grown under WW and DS conditions. Orthogonal partial least squares discriminant analysis (OPLS-DA) indicated that 32.5% of the metabolite features contributed to drought response (VIP ≥ 1, 1267/3890). Paired *t*-test showed that most of the metabolite features were significantly influenced by drought stress, with 37.4% (1301/3483) showing notable changes in accumulation (|Fold Change| ≥ 2, false discovery rate [FDR] < 0.05). In total, 1035 metabolites were detected as drought-responsive metabolites from the intersection part of the results based on the above assays (Fig. 1c; Additional file 1, Figure S1b; Additional file 2, Table S3). Notably, most (83.3%, or 862/1035) metabolites showed upregulation patterns in response to drought (Fig. 1c), indicating that upregulation of genome-wide metabolites tends to have dominant roles in control of maize drought response, which might confer plant drought tolerance. Of these metabolites, 71 were annotated and belonged to broad range of classes, including phenolamides, lipids, sugars, amino acids, benzoxazinoids, polyphenols, organic acids, nucleic acids, flavonoids, and vitamins (Fig. 1d; Additional file 2, Table S2). Interestingly, the drought-responsive metabolites from the phenolamide, lipid, sugar, benzoxazinoids, and nucleic acid families only showed upregulation patterns, while drought-responsive vitamins were only downregulated (Fig. 1d).

Previous studies have reported the drought-tolerant phenotype survival rate of most of our 385 inbred lines (Additional file 2, Table S1) [9]. We further investigated dry masses and fresh weights as drought-tolerant indices of the 385 inbred lines in drought experiments (Additional file 2, Table S1, see "Materials and methods"). Based on these drought-tolerant indices, we performed maize drought tolerance prediction with 1035 drought-responsive metabolites, 3890 total metabolites, and 1035 randomly selected ones via rrBLUP approach (see "Materials and methods") [25]. The results showed that the prediction accuracy for drought tolerance, especially for survival rate, using 1035 drought-responsive metabolites, was significantly higher than those using the total or randomly selected metabolites (Fig. 1e). In addition, we found that a combination of 15 m-traits could explain more than 60% of the phenotypic variance of survival rate (Additional file 1, Figure S2; Additional file 2, Table S4). Together, these data suggested that the drought-responsive metabolites could efficiently reflect maize response and tolerance to drought, and our subsequent analyses were focused on the 1035 drought-responsive metabolites accordingly.

### Genetic basis of the metabolome variations underlying maize drought response

Previous studies have reported ~ 1.25 M single-nucleotide polymorphism (SNP) markers of the maize lines used in this study [24]. To unravel how the drought-responsive metabolites are genetically controlled, we performed GWAS to detect significant associations between SNP markers and the drought-responsive metabolites (or m-traits) with mixed linear model (MLM) controlling population structure (Q) and

kinship (K) using TASSEL5 (https://tassel.bitbucket.io/). Around 88% (921/1035) of the m-traits had at least one significantly associated locus (MLM, $N = 385$, $P < 2.1 \times 10^{-6}$). In total, 7811 distinct significant SNPs associated with 921 m-traits were detected (Fig. 2a; Additional file 2, Table S5). More significant SNPs (50.6% or 3951/7811) were detected by GWAS under DS condition as compared to those under WW condition (38.7%, or 2955/7811), and only 11.6% (905/7811) of the SNPs were statically detected under both conditions (Additional file 1, Figure S3a), indicating that the m-trait-SNP associations were very sensitive to the growth environment. Each SNP marker explained 5.9 ~ 35.8% (median, 7.8%) of the observed phenotypic variance of the m-traits (Additional file 1, Figure S3b). After significant SNPs within 10 kb flanking region were merged, a total of 3415 loci were identified as mQTLs (Additional file 2, Table S6). 21.4% (731/3415) of these QTLs showed co-localization with QTLs reported previously in maize drought studies based on the

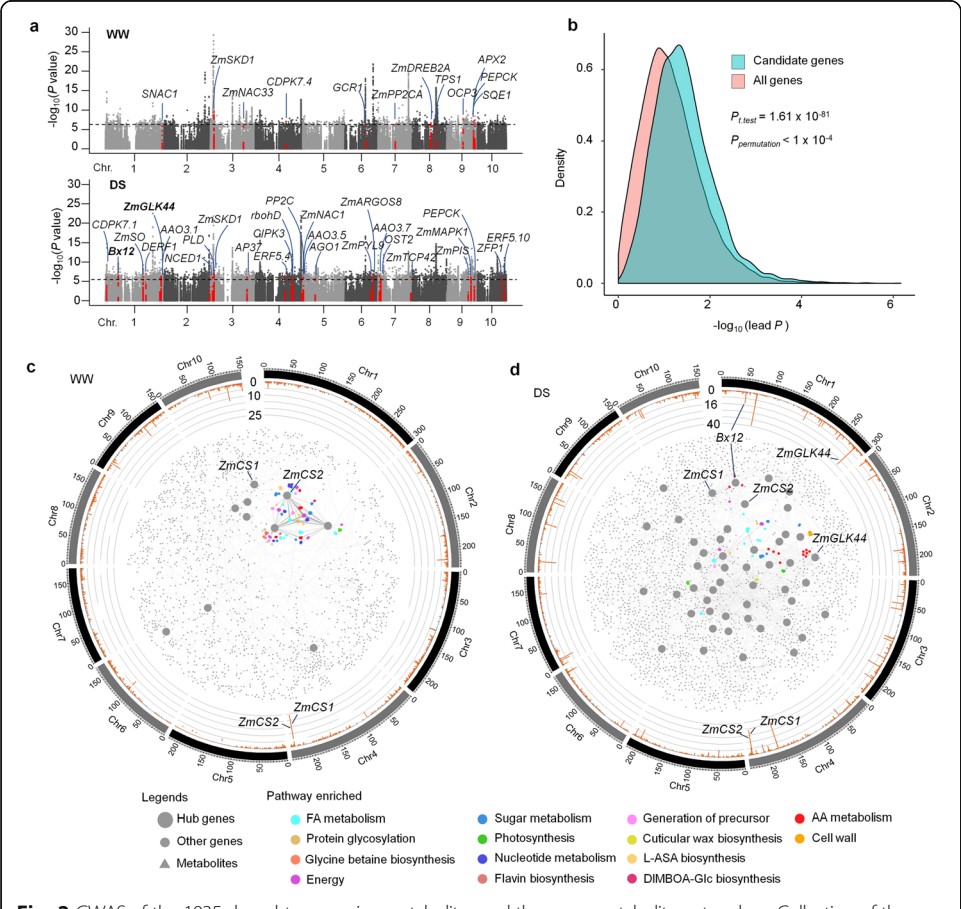

**Fig. 2** GWAS of the 1035 drought-responsive metabolites and the gene-metabolite networks. **a** Collection of the GWAS signals for the 1305 drought-responsive metabolites detected under WW (up panel) and DS (bottom panel) conditions. The dashed lines indicate the threshold of significance. The loci without significant SNPs were filtered out. All the 35 candidate genes with known drought tolerance roles were annotated based on the significant GWAS signals and indicated in the panels. **b** Density plot showing more significant SNPs associated with maize drought tolerance in the candidate genes as compared to the same number of randomly selected genes. The significance was calculated based on 1000 times of permutation test. **c, d** Gene-metabolite networks using 1035 drought-responsive metabolites and 2589 candidate genes under WW (**c**) and DS (**d**) conditions. Circles showing the chromosomal distribution of the genes associated with various numbers of metabolites under WW (**c**) and DS (**d**) conditions. The imbedded networks show the gene-metabolite associations or correlations under WW (**c**) and DS (**d**) conditions. Two hub genes *ZmCS1* and ZmCS2 detected under both WW and DS conditions are indicated while two other hub genes Bx12 and ZmGLK44 detected only under DS conditions are indicated

traditional drought response indices (Additional file 2, Table S6) [26, 27]. Therefore, our m-trait-based GWAS could detect many new putative drought-related QTLs in addition to those of known.

GWAS mapping with this maize population was able to reach single-gene resolution [23]. Based on the peak SNPs (SNP with smallest $P$ value at a locus) and their linkage disequilibrium (LD), a total of 2589 candidate genes were identified based on the 7811 significant SNPs (Additional file 2, Table S5). In total, 190 genes among them were TFs (Additional file 2, Table S5), which showed a significantly enrichment as compared to the total genome-wide TFs ($p = 6.6 \times 10^{-4}$, Fisher's exact test). These results suggested the importance of transcriptional variations in maize drought tolerance regulation. The candidate genes were more significantly enriched under DS condition than WW condition in several Kyoto Encyclopedia of Genes and Genomes (KEGG) pathways such as "TCA cycle," "Glycolysis / Gluconeogenesis," and "Auxin signaling" (Additional file 1, Figure S4a-d; Additional file 2, Table S7), indicating that these enriched variations may contribute to the divergence of the maize drought tolerance by regulating these pathways under DS condition.

Moreover, thirty-five known drought-tolerant genes were included in the candidate gene set (Fig. 2a; Additional file 2, Table S8). For example, *ZmSKD1* (*suppressor of K⁺ transport growth defect 1*), an AAA-type ATPase encoding gene, was previously reported to play a positive role in drought tolerance [28]. Twelve SNPs of *ZmSKD1*, which were in the same LD block ($R^2 > 0.2$), were significantly associated with metabolite PN_Group_00629 under both WW and DS conditions (Additional file 1, Figure S5a-d). There were two alleles, C and T, of the significant SNP chr3.S_19462144. Drought promoted the production of PN_Group_00629 in plants with both alleles, but less PN_Group_00629 was produced and a higher survival rate was observed in plants with allele C than in plants with allele T (Additional file 1, Figure S5e and f). Therefore, allele T could be a favorable allele that downregulated PN_Group_00629 but promoted maize drought tolerance. Further, we performed association analysis between survival rate and the candidate genes. Randomly selected genome-wide genes were used as control. The results showed significant enrichment of candidate gene SNPs as compared to those of random genes (Fig. 2b; Additional file 2, Table S5) suggesting that the survival rate-gene associations were not false positive and that they were likely involved in regulation of maize drought tolerance. Next, we ranked the genes based on the $p$ value of the peak SNP of their locus. In the top 100 genes with most significant SNPs, 19 of them were found in our candidate gene set, while the expectation gene number is 8 (Fisher's exact test, $p = 6.4 \times 10^{-4}$). The number of candidate gene in top 500 gene list is 72, while the expectation gene number is 41 (Fisher's exact test, $p = 3.25 \times 10^{-6}$), and we observed 126 genes in top 1000 gene list, while the expected genes are 82 (Fisher's exact test, $p = 1.35 \times 10^{-6}$). These results indicated that our candidate genes might be enriched in the putative drought-tolerant genes.

The transcriptomes of 197 maize natural inbred lines (a core germplasm of the maize population used in the metabolome profiling) grown under WW and DS conditions have been generated (Dai M, unpublished). Based on the centrality-lethality rule [29], correlations were calculated between the 1035 drought-responsive metabolites and gene expression under WW and DS conditions. In total, 4787 metabolite-gene correlations ($p < 1 \times 10^{-5}$) were detected, with 818 metabolites and 2873 genes involved (Additional

file 2, Table S9). Of all these metabolite-associated or metabolite-correlated genes, 56 were hub genes (see "Materials and methods"), most of which (82%, or 46/56) were specifically identified under the DS condition (Fig. 2c, d; Additional file 2, Table S10). Three hub genes *ZmCS1* (*citrate synthase 1*, GRMZM2G063909), *ZmCS2* (*citrate synthase 2*, GRMZM2G064023), which are two putative citrate synthase genes [30] (Additional file 1, Figure S6), and *ZmMYBR88* (GRMZM2G064328) were statically detected under both WW and DS conditions. Previous studies have reported the associations of *ZmCS1* with phenylpropanoid hydroxycitric acids [30], which were also identified in current study (Additional file 1, Figure S7a; Additional file 2, Table S5). In addition, we detected significant associations between *ZmCS1* and many other metabolites such as flavonoids, carboxylic acid, benzoxazinoids, and others (Additional file 1, Figure S7a-c). We simultaneously detected *ZmCS2* (~ 10 kb next to *ZmCS1*) that was associated with these metabolites apart from *ZmCS1* (Additional file 1, Figure S7a-c). Between *ZmCS1* and *ZmCS2*, only the expression of *ZmCS2* was upregulated after DS ($P < 9.28 \times 10^{-11}$, paired *t*-test), and the associations between metabolites and *ZmCS2* were more sensitive to drought than those between metabolites and *ZmCS1* (Additional file 1, Figure S7d and e).

Of these hub genes, *ZmPP2C-A10* and *ZmVPP1* are known to play roles in the maize drought tolerance [9, 10]. These data again demonstrated the strong power and reliability of our analyses in mapping causative genes that regulate maize metabolites in the maize drought response and tolerance. Further, genes co-expressed ($P < 4.67 \times 10^{-9}$) with the hub genes were identified. Pathways (CornCyc, https://corncyc-b73-v3.maizegdb.org/) enriched with the co-expressed genes were detected (Fig. 2c, d). These pathways helped in the functional interpretation of the hub genes in context of metabolic regulation. For example, genes co-expressed with the hub genes *Bx12* (GRMZM2G023325, encoding an enzyme for the benzoxazinoid metabolic pathway) and *ZmGLK44* (GRMZM2G124540, encoding Golden 2-like TF) identified under DS condition were enriched in DIMBOA-Glc biosynthesis and amino acid metabolism (Fig. 2d), suggesting important roles of *Bx12* and *ZmGLK44* in regulating these metabolic pathways in maize drought response.

### Natural variants that regulated the expression divergences of the candidate genes

Based on the transcriptome of 197 natural inbred lines grown under WW and DS conditions, we investigated the expression QTLs (eQTLs) that regulated the expression of the 2589 candidate genes associated with drought-responsive metabolites. In total, 13,546 distinct eQTLs (MLM, $P < 4.2 \times 10^{-8}$) were identified for 54.8% (1421/2589) of the candidate genes (Additional file 2, Table S11; Additional file 2, Table S11-13), suggesting that transcription variations could play important roles in regulation of metabolites of maize population in drought response or tolerance. An eQTL was defined as a *cis* eQTL if the lead SNP was located in the 20 kb up or downstream region of an expression trait gene (e-trait gene), but otherwise it was a *trans* eQTL [31]. Of the 13,546 eQTL, the majority (91.1%, or 12,343/13,546) were *trans* eQTLs (Fig. 3a; Additional file 2, Table S13), and the remaining (8.9%) were *cis* eQTLs. These data indicated that there existed complex regulatory networks involved in drought responses or tolerance of the

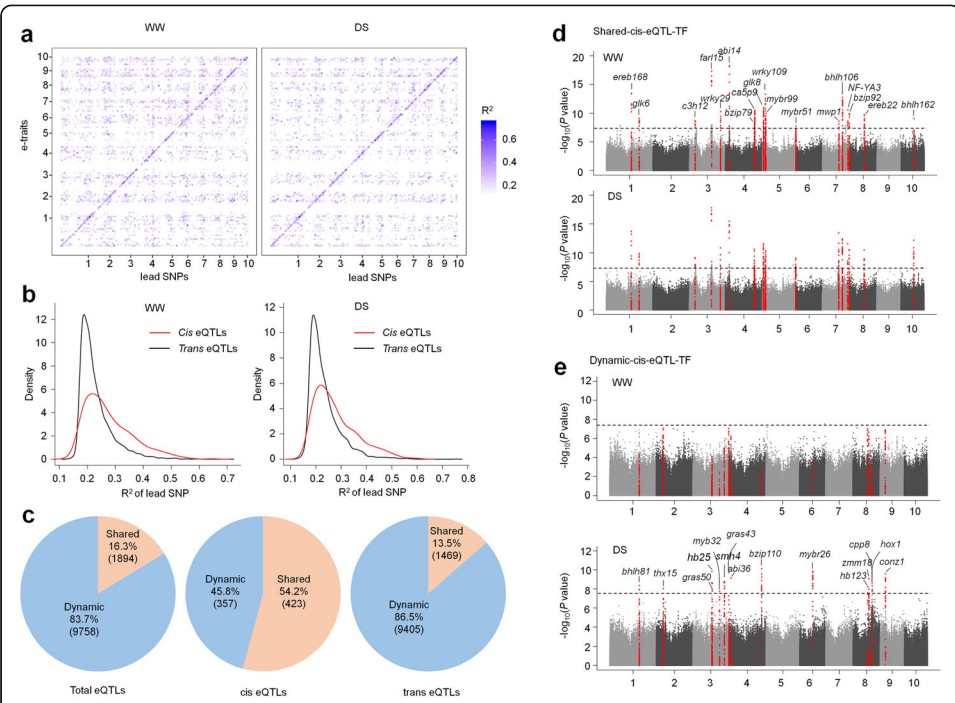

**Fig. 3** eQTLs that control the expression of the candidate genes. **a** The distribution of local and distant significant SNP markers that are associated with the candidate gene expression under WW and DS conditions. **b** Density plot showing the explained expression trait (e-trait) variance by each significant SNPs under WW and DS condition. **c** The dynamic and shared eQTLs in total, *trans* and *cis* eQTL groups. **d, e** The local eQTLs that were located in TF genes detected under both WW and DS conditions (static, **d**) or only under DS conditions (dynamic, **e**)

maize population. However, the *cis* eQTLs generally explained larger phenotypic variations of the expression traits as compared to the *trans* eQTLs (Fig. 3b).

Most of the eQTLs (84%) were dynamic (identified under either WW or DS growth conditions), only 16.3% were shared (identified under both growth conditions). Similar ratios of dynamic and shared eQTLs were observed for *trans* eQTLs (Fig. 3c). Interestingly, there were more shared *cis* eQTLs (54.2%) in contrast with the dynamic ones (45.8%) (Fig. 3c). We looked more deeply into the *cis* eQTLs as they, on average, had larger effects than *trans* eQTLs on e-trait gene expression regulation (Fig. 3b). For example, some *cis* eQTLs were stably and repeatedly detected in genes response to osmotic stress (*ara1, rplP10, ZmSLT1, ZmFes1A, ZmRABG3c, ZmSnRK3.16, ZmAM-MECR1, ZmNADP-QD, ZmPIMT1, ZmBAG4, ZmCOBL1*), glucan metabolism (*ZmSS2, ZmDsPTP1, ae1, ZmABI8, ZmXTH30*), and cytoskeleton organization (*ZmCAP1, ZmTCTP, ZmFACb, ZmMPK20*) (Additional file 2, Table S12). In addition, there were many *cis* eQTLs being involved in regulating the expression of e-trait TF genes (*abi14, bhlh106, bhlh162, bzip79, bzip92, c3h12, ca5p9, ereb22, ereb168, farl15, glk6, glk8, myb99, mybr51, mwp1, NF-YA3, wrky29, wrky109*) (Fig. 3d). These data suggested strong effects of *cis* eQTLs on the expression regulation of the genes nearby. For the dynamic *cis* eQTLs, significant and unique peaks were identified under DS conditions for genes involved in benzoxazinoids synthesis (*Bx12*), cellular response to stress (*gst19, mmp165, ZmGR1, ZmRBP-DR1, ZmRECA3, ZmMCB1*), ABA response (*ZmLEW3, ZmMCB1, ZmSnRK3.17*), phenylpropanoid hydroxycitric acid (*ZmCS1, ZmCS2*), and

transcription regulation (*abi36, bhlh81, bzip110, conz1, cpp8, gras43, gras50, hb25, hb123, hox1, mybr26, myb32, smh4, thx15, zmm18*) (Fig. 3e; Additional file 2, Table S12). Therefore, these *cis* variants could be more specifically involved in stress response of these e-trait genes in the maize population.

### *Bx12* contributing to the divergence of maize drought tolerance via regulating DIMBOA-Glc biosynthesis

In this study, we identified a diversity of new candidate genes associated with metabolite contents in maize drought responses by experimentally characterizing two hub candidate genes, *Bx12* and *ZmGLK44*, for the purpose of showing validation in principle is possible and providing new insights of molecular mechanisms underlying metabolome-mediated maize drought tolerance.

The grass-specific benzoxazinoids play important roles in pest defense and disease resistance [32], but whether and how these metabolites are involved in plant abiotic stress tolerance remain elusive. Various benzoxazinoids, which are catalyzed by different *Bx* and *GLU* genes, were detected in maize leaves (Fig. 4a) [32]. In our study, we detected 18 different benzoxazinoids in maize plants grown under WW and DS conditions (Additional file 2, Table S2). Eight of these benzoxazinoids and 7 other drought-responsive metabolites were significantly associated with SNPs in the region of 66–67 Mb on chromosome 1 (Additional file 2, Table S5). Conserved and dynamic associations in this region were observed based on different metabolites. For instance, some SNPs were significantly in association with HBOA contents under both WW and DS conditions, but regarding DIMBOA-Glc contents only under DS conditions ($P < 2.1 \times 10^{-6}$, $N = 385$), showing two peaks of significant SNPs located in *Bx11* (GRMZM2G336824) and *Bx12* loci (Fig. 4b–e; Additional file 1, Figure S8a). As a product of DIMBOA-Glc, HDMBOA-Glc also showed significant association with *Bx12* loci no matter with or without drought stress (Additional file 1, Figure S8b). Interestingly, HDMBOA-Glc was not a drought-responsive metabolite based on our standard. The significant SNPs from the two peaks showed high linkage ($R^2 > 0.51$) to each other and were located in the same LD block (Additional file 1, Figure S8c-h). Previous reports have shown that an InDel of CACTA TE (TE_InDel) in *Bx12* is the causal variant that controls the divergence of *Bx12* expression and further DIMBOA-Glc production in maize leaves [33], which is in accordance with our results (Fig. 4f, g). Moreover, we observed that *Bx12* expression was enhanced in the maize population after drought stress (Fig. 4f, g; Additional file 2, Table S11), suggesting potential roles of *Bx12* in maize drought tolerance.

Next, we focused on the CACTA TE to investigate how *Bx12* functions in maize drought tolerance. Among the 385 maize lines, plants with a TE deletion allele had significantly higher survival rate after drought stress than those with a TE insertion allele (Fig. 4h; Additional file 3, Table S14). We further generated three maize linkage populations, two $F_2$ populations derived from CIMBL144 ($TE^{+/+}$) × CIMBL55 ($TE^{-/-}$) and B73 ($TE^{+/+}$) × CIMBL70 ($TE^{-/-}$), and a heterogeneous inbred family (HIF) population based on the CACTA TE at the *Bx12* locus from the previously reported recombinant inbred line (RIL) populations derived from a cross between modern maize Zong3 ($TE^{+/+}$) and the maize ancestor teosinte ($TE^{-/-}$) [34] to verify this observation. All these three populations showed significantly higher survival rates for plants with a TE deletion allele than for ones with a TE insertion allele (Fig. 4i–m), suggesting that the TE

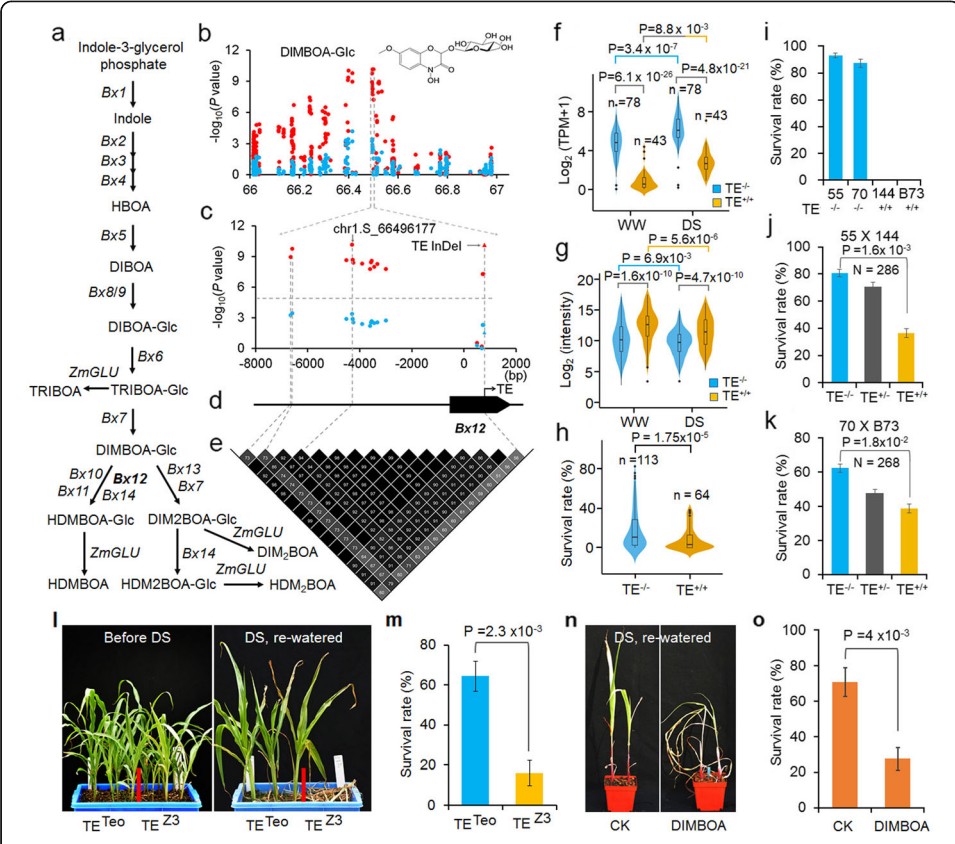

**Fig. 4** Association of *Bx12* locus with DIMBOA-Glc levels and survival rates. **a** The biosynthetic pathway of benzoxazinoids. Bx12 and other Bx and ZmGLU enzymes are indicated in the pathway. **b, c** Overview (**b**) and closer view (**c**) of the associated SNPs in the region Chr1 66-67 Mb where *Bx12* locus was located. **d** Gene model of *Bx12*. The CACTA TE located in *Bx12* is indicated. **e** The LDs among the significant SNPs of *Bx12*. **f, g** The *Bx12* expression levels in the 197 core inbred lines (**f**) and the DIMBOA-Glc levels in the 385 inbred lines (**g**) with (TE$^{+/+}$) or without (TE$^{-/-}$) TE insertion under WW and DS conditions. **h** Survival rates of the maize inbred lines after drought stress. **c** Survival rates of maize inbred lines CIMBL55 (55), CIMBL70 (70), CIMBL144 (144) and B73 after drought stress. **j, k** Survival rates of segregated F$_2$ plants derived from the cross 55 × 144 (**j**) or 70 × B73 (**k**) after drought stress. Error bars: s.e, based on three biological repeats; statistical significance was determined by Student's *t* test. **l** Growth comparison of the HIF plants derived from Zong3 (TE$^{+/+}$) × Teosinte (TE$^{-/-}$) before (upper panel) and after drought (bottom panel). **m** Survival rates of HIF plants exposed to drought and re-watered conditions respectively. Error bars, s.e. based on three biological repeats (n = 6); statistical significance was determined by paired *t.*test. **n, o** The growth (**n**) and the survival rate (**o**) of maize B73 plants after drought-stressed (DS) with or without (CK) DIMBOA treatment. Error bars, s.d. based on 6 biological repeats (n = 24); statistical significance was determined by paired *t* test.

deletion allele is favorable for drought tolerance. DIMBOA-Glc is the stock form of the active form DIMBOA in the plant cells [32]. Considering both DIMBOA-Glc and DIMBOA have very similar effects on cell responses [35, 36], we used the commercial chemical DIMBOA to evaluate the role of DIMBOA-Glc in maize drought response. Spraying DIMBOA to the maize genotype B73 decreased the survival rates of maize plants after drought treatment, while DIMBOA treatment showed no significant effect on plants grown under well-watered conditions, indicating negative roles of DIMBOA and DIMBOA-Glc in maize drought tolerance (Fig. 4n, o; Additional file 1, Figure S9). Together, these data suggested that *Bx12* played positive roles in maize drought tolerance likely through downregulation of DIMBOA-Glc accumulation upon drought stress.

Previous studies have indicated that the TE insertion in *Bx12* underwent selection during maize domestication [37]. We further investigated the Pre-Columbian distribution of this TE in the Americas with a published landrace population [38]. The frequencies of this TE in landraces from the USA were significantly higher than those in landraces from the other regions of Americas (Additional file 1, Figure S10a and b; Additional file 3, Table S15). In addition, the TE insertion allele was significantly associated with higher latitudes (Additional file 1, Figure S10c; Additional file 3, Table S15). These data indicated that the TE insertion in *Bx12* underwent strong selection during the spread of maize into the US temperate zone. Given the increased tolerance of maize plants with TE deletion allele than with insertion allele, we propose that there could be a selection balance between drought tolerance and other agronomic traits associated with the *Bx12* locus during maize domestication and spread.

### *ZmGLK44* regulates tryptophan accumulation via activating tryptophan biosynthesis gene expression

Golden 2 (G2) and its homologous Golden 2-like (GLK) TFs are plant-specific TFs, belonging to the GARP subfamily of MYB TFs [39, 40]. GLK TFs are vital to plant development [39, 41]. Nonetheless, how these TFs function in plant drought tolerance remains unknown. Based on our data sets, *ZmGLK44* is a hub gene correlated / associated with 30 drought-responsive metabolites (Fig. 5a; Additional file 2, Table S10). Eight of these are annotated amino acids (Fig. 5a; Additional file 2, Table S2). To elucidate the roles of *ZmGLK44* in regulating the biosynthesis of these drought-responsive metabolites, we generated *ZmGLK44* inducible overexpression transgene maize lines (Additional file 1, Figure S11a and b). Metabolic profiling of these transgenic plants and their negative siblings showed that most (60%, or 18/30) of the putatively targeted metabolites were not significantly changed under WW condition, but upregulated to higher levels in positive transgenic plants than in their negative siblings under DS condition (Fig. 5b; Additional file 3, Table S16). The upregulated metabolites included tryptophan (Trp) and two tryptophan-related metabolites achillamide derivatives. We next performed RNA-seq to *ZmGLK44*-positive and *ZmGLK44*-negative transgenic plants grown under DS condition. The differentially expressed genes were enriched in several metabolic pathways, including "glycerophosphodiester degradation," "L-tryptophan biosynthesis," and "DIBOA-glucoside biosynthesis," (Fig. 5c Additional file 1, Figure S12 a-c; Additional file 3, Table S17) [42], suggesting a role of *ZmGLK44* in regulation of these metabolic pathways.

Both metabolic profiling of the 385 inbred lines and RNA-seq of the transgenic plants indicated a role of *ZmGLK44* in regulation of Trp accumulation. We therefore investigated how *ZmGLK44* regulates Trp biosynthesis. Three key genes—*ASA2*, *PAT1*, and *TSB2*—that encode synthetic enzymes of the Trp biosynthesis pathway, were upregulated in the positive transgenic plants as compared to the negative siblings (Fig. 5d, e). Consistently, the metabolites indole and Trp were increased to significantly higher levels in positive transgenic plants compared to the negative siblings after DS (Fig. 5f). GLK TFs bind to a *cis* element with a CCAATC core sequence [39]. To know how ZmGLK44 regulates *ASA2*, *PAT1*, and *TSB2*, the fragments *ASA2p*, *PAT1p*, *TSB2p1*, and *TSB2p2* with CCAATC elements in *ASA2*, *PAT1*, and *TSB2* promoters,

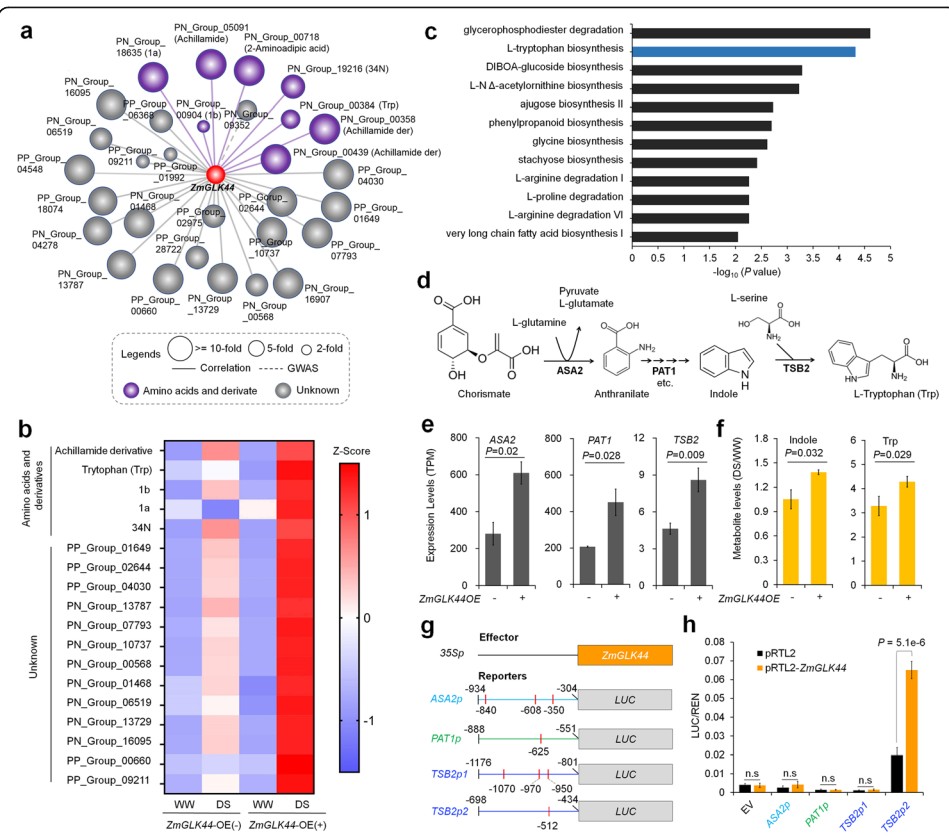

**Fig. 5** *ZmGLK44* regulates Trp biosynthesis via activating the genes encoding enzymes involved in Trp biosynthetic pathway. **a** Metabolites correlated/associated with *ZmGLK44*. The size of circles represents fold changes of metabolite levels in drought response. Metabolites annotated as amino acid derivatives were indicated as purple circles. 1a: 1-Acetyl-3-carboxy-3,4-dihydro-β-carboline; 1b: ethyl 6-acetamido-4-hydroxyquionline-3-carboxylate; 34 N: N-Acetyl-beta-oxotryptamine. **b** Accumulation of *ZmGLK44* correlated/associated metabolites in *ZmGLK44* transgenic plants and their negative siblings grown under WW and DS conditions. **c** Pathways with enriched DE genes. L-tryptophan (Trp) biosynthesis pathway is highlighted. **d** Main enzymes (*ASA2*-GRMZM2G161337, *PAT1*-GRMZM2G051219, and *TSB2*-GRMZM2G005024) and metabolites involved in Trp biosynthesis process. **e** Expression of *ASA2*, *PAT1*, and *TSB2* in *ZmGLK44* transgenic plants and their negative siblings grown under DS conditions. **f** After DS, Indole, and Trp were enhanced in production in positive transgenic plants as compared to the negative siblings. **g** Diagram showing the effector (35S:*ZmGLK44*) and reporters (*Asa2p:LUC*, *PAT1p::LUC* and *PAT1p::LUC*) used in protoplast assays. The short-red lines indicate the putative GLK TF binding sites and their location in the promoters of each gene. The numbers indicate the positions relative to the translation starting site ATG (+ 1) of each gene. **h** Activation of *TSB2p2:LUC*, but not the other promoter fragments, by ZmGLK44 in maize protoplasts. Error bars: s.d, based on three biological repeats; statistical significance was determined by Student's *t* test; n.s., not significant

respectively, were cloned and used in LUC assays (Fig. 5g). In maize protoplasts, enhanced expression of *ZmGLK44* increased the reporter LUC expression levels driven by the fragment *TSB2p2*, but not *ASA2p*, *PAT1p*, nor *TSB2p1* (Fig. 5g, h). These data indicated that *TSB2* could be a direct target of ZmGLK44 and that the flanking sequences of CCAATC could be important for the binding of GLK TFs to the core cis element.

### *ZmGLK44* positively regulated maize drought tolerance

Trp is an aromatic amino acid showing responses to water-deficient/drought stresses from very early to late stages [11]. Spraying Trp onto different maize genotypes increased the survival rates of maize plants after drought treatment (Fig. 6a–d), indicating

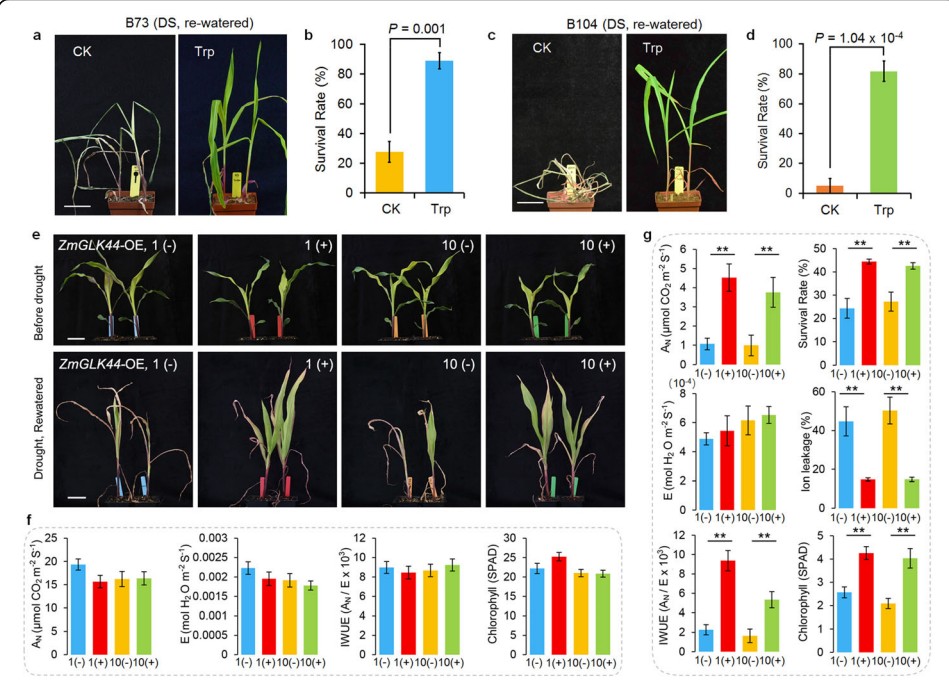

**Fig. 6** *ZmGLK44* and Trp positively regulated maize drought tolerance. **a–d** The growth of B73 (**a**) and B104 (**c**) plants before Trp treatments (upper panels) and after Trp treatments followed by drought stress (bottom panels), and survival rates of Trp-treated B73 (**b**) and B104 (**d**) plants grown under DS conditions. The control treatments (CKs) were sprayed with 0.1% tween-20. **e** Growth of positive (+) and negative (−) *ZmGLK44* transgenic plants before drought stress (upper panels) and re-watered after drought stress (lower panels). Two independent transgenic lines #1 and #10 are illustrated. **f** Photosynthetic rates ($A_N$), transpiration rates (E), water use efficiencies (WUE, $A_N$/E), and chlorophyll of the transgenic lines #1 and #10 without drought stress. **g** $A_N$, E, WUE ($A_N$/E), chlorophyll, ion leakage, and survival rates of transgenic lines #1 and #10 after drought stress. Error bars in **f** and **g**: s.e average stand deviations based on biological repeats ($n$ = 9); statistical significance was determined by a two-sided $t$ test: *$P$ < 0.05, **$P$ < 0.01

a positive role of Trp promoting maize survival upon drought. Given the higher in-creased levels of Trp in *ZmGLK44* positive transgenic plants compared to the negative siblings after drought stress (Fig. 5f), we hypothesized that *ZmGLK44* could function positively in maize drought tolerance. To test this hypothesis, both positive transgenic plants and negative siblings were grown in the soil under WW and DS conditions (Fig. 6e). Under WW condition, the indices of $CO_2$ assimilation ($A_N$), $H_2O$ transpiration (E), instantaneous water use efficiency (WUE, $A_N$/E), and chlorophyll made no difference between positive plants and their negative siblings (Fig. 6f). However, following drought treatment, the levels of $A_N$, WUE, and chlorophyll, but not E, were significantly higher, while the levels of ion leakage were significantly lower in the positive plants than the negative siblings (Fig. 6g). As a result, the positive plants had higher survival rates after drought stress (Fig. 6e, g). Taken together, these results demonstrated a positive role of *ZmGLK44* in maize drought tolerance.

## Discussion

In recent years, many metabolomes and their genetic control have been uncovered in a variety of crops [30, 43–45]. While crops grown under normal conditions have been mostly studied, the research in this field has not revealed the population-level metabo-lomes involved in crop environmental stress responses and their genetic control. Appli-cation of high-throughput metabolite profiling to the 385 natural inbred lines grown

under WW and DS conditions resulted in the identification of more than one thousand drought-responsive metabolites, most of which were upregulated. The drought-responsive metabolites identified in this study represent the hitherto largest body of internal responses of large crop populations to drought, exacerbating our study's potential contribution to drought studies in other crops. Based on GWAS, we detected more genetic loci associated with these drought-responsive metabolites under the DS condition than those under the WW condition. In addition, there were very few overlapping loci between these two groups. Therefore, the drought-responsive metabolome is subject to substantially distinct genetic regulations under different environments.

We detected more than three thousand mQTLs based on the significant associations. Aside from overlapping QTLs as compared to the previously identified ones using traditional drought phenotypes [6, 26, 27, 46], there were a large proportion of the mQTLs (~ 78.6%) not detected in previous studies, pointing at the possible bottlenecks of traditional traits and the power of m-traits in mapping drought-responsive QTLs. We observed a vast amount of large-effect mQTLs associated with the natural variations of secondary metabolites in drought response as compared to the QTLs detected for the traditional drought-related traits [6, 26, 27, 46]. This might be attributed to the lower complexity of the genetic architectures of the m-traits in drought response, or to the highly enriched diversities of these m-traits as compared to the traditional traits. Unlike GWAS or linkage mapping with traditional traits, among which there is no mutual enhancement for functional annotation of candidate genes, the eQTL-gene-metabolite interaction networks in this study were greatly beneficial for functional gene annotation and provided novel insights into the biochemical and regulatory basis of crop drought response.

In addition to those known metabolites to play roles in abiotic stress responses, we detected many metabolites, including benzoxazinoids which were of particular interest, with unknown roles in drought response. Benzoxazinoids are plant-specific metabolites, mainly produced in cultivated or wild Poaceae, such as wheat, rye, maize, and its ancestor teosinte [37, 47]. Previous studies have found that benzoxazinoids are dramatically induced by insect herbivory [48, 49]. In addition, they can also be induced by fungal pathogens infection and by jasmonic acid treatment [50–53]. In fact, benzoxazinoids represent the main endogenous chemicals that are of great significance to insect and pathogen defense responses [32]. The roles of benzoxazinoids in abiotic stress responses or tolerance remain largely unknown. The natural variation of *Bx12* was significantly associated with the levels under drought stress of DIMBOA-Glc, a prominent benzoxazinoid and a direct substrate of Bx12 [32]. We observed increased expression levels of *Bx12* alongside reduced levels of DIMBOA-Glc in maize plants grown under DS condition as compared to plants grown in WW (Fig. 4). In addition, the expression of *Bx12* was positively correlated with maize survival rates after drought stress. Therefore, *Bx12* likely plays a positive role in maize drought tolerance via regulation of DIMBOA-Glc levels, suggesting a new role of benzoxazinoids in plant abiotic stress tolerance other than defense responses. Feeding by lepidopteran species can induce DIMBOA-Glc methylation to form HDMBOA-Glc, which was shown to enhance maize resistance to insect herbivores [32]. Given that *Bx12* is directly involved in conversion of DIMBOA-Glc to HDMBOA-Glc and positively regulates maize drought tolerance (Fig. 4), *Bx12* and its natural variation CACTA TE could be potential genetic resources or molecular markers in breeding maize cultivars with both drought tolerance and lepidopteran species resistance.

In total, 190 of the candidate genes were TFs, representing 7.5% (190/2538) of the total TFs in the maize genome [54]. This indicates that transcription regulation contributes greatly to the natural metabolic variations in maize drought response. We validated one of these TF genes, *ZmGLK44*, which was also a hub gene associated with dozens of metabolites, with regard to regulation of Trp biosynthesis and further maize drought tolerance, providing new insights regarding the involvement of GLK TFs into environmental stress adaptation. After drought induction, *ZmGLK44* overexpression in transgenic maize enhanced photosynthetic rates, chlorophyll, and further water use efficiency, indicating a role of ZmGLK44-controlled Trp in regulation of photosynthetic pathways.

ZmGLK44 also regulated the expression of genes *ZmPYL12*, *ZmSnRK2.1*, *SnRK2.4*, *ZmPP2C-A4*, *ZmPP2C-A7*, *ZmRD17*, *ZmRAB18*, and *ZmLEA* that are involved in ABA signaling (Additional file 1, Figure S13), which plays key roles in plant drought response and tolerance [55]. Given the key roles of ABA in environmental stress tolerance, we deduced that ZmGLK44 could promote the Trp levels through activating the expression of Trp biosynthetic genes, which further enhanced ABA signaling and water use efficiency to help plants in resisting drought. A recent study has suggested a role of maize GLK TFs in promoting crop yield [41]. Therefore, *ZmGLK44* could be potentially used in breeding new maize cultivars with both increased yield and drought tolerance.

In this study, we detected a total of 1035 drought-responsive metabolites. More than 60% of the phenotypic variance of survival rate could be predicted by the combined 15 m-traits (Additional file 1, Figure S2; Additional file 2, Table S4), indicating that these m-traits could be used as biomarkers to select maize drought-tolerant germplasms. M-trait-based GWAS resulted in the identification of more than two thousand candidate drought-responsive genes. A large number of genes showed significant enrichment in the KEGG pathways, most of which enriched in these pathways were upregulated in the 197 maize inbred lines after drought stress, like *ZmCS2*, *ZmACO3*, *ZmclCDH*, *ZmMDH* in TCA cycle, *ZmHXK1*, *ZmPFK3*, *ZmTPI*, *ZmGAPC1*, *ZmIOGAM1 / 2*, *ZmPK2 / 3*, *ZmPDC3*, *ZmAKR4C10.2* in glycolysis / gluconeogenesis, and *ZmTIR1*, *ZmARF3.1 / 3.2* in auxin signaling (Additional file 1, Figure S4). Most of these upregulated genes exert influences upon regulation of stress responses, especially *ZmHXK1*, *ZmGAPC1*, *ZmPDC1*, *ZmALDH4*, and *ZmARF3.1 /3.2*, whose orthologs in other species have been reported to positively regulate plant drought tolerance [56–60]. Notably, we validated the possible roles of the candidate genes in metabolite biosynthesis and drought tolerance using several approaches, including association analysis with survival rates, studies in linkage populations, and transgenic lines subjected to comprehensive molecular and biochemical assays. Considering that most loci mapped in this study are novel drought-responsive loci, our results thus provided reliable and rich materials for further studies of plant stress responses.

## Conclusions

From its ancestor teosinte to landraces, maize underwent domestication 9000 years ago [61]. Many agronomic traits, especially those involved in yield increase, have been further improved during the modern maize breeding processes. Thousands of genes have been involved in the domestication and improvement processes [62] which lead to the

exclusion of partial drought-tolerant genes or alleles whereas the drought sensitivities of modern maize cultivars have steadily increased during the breeding selections [2] (Additional file 1, Figure S14). Importantly, drought may have a much more severe effect on the yield loss of maize than what model predictions allow [2]. Due to genetic bottlenecks, which could result from the loss of stress-tolerant genes or from favorable variations during domestication and improvement, e.g., depletion of drought-tolerant genes as revealed in this study (Additional file 1, Figure S14), re-domestication and precise genetic design could be necessary to develop novel crop cultivars with both high yield and stress tolerance, thus to meet the increased yield demands and environmental sustainability [63]. The candidate drought-tolerant genes and the related natural variations detected in our study could serve as invaluable resources for breeding those novel cultivars.

## Materials and methods

### Plant materials and growth conditions

The 385 maize inbred lines were sown and grown in two growth pools at Huazhong Agricultural University, Wuhan, China (Additional file 1, Figure S1a). The pools were covered with soil at a height of 25 cm. It has been reported that maize plants grown in the field with soil moisture (SM) of 25% have normal transpiration and photosynthesis rates, while the plants are drought-stressed with greatly reduced transpiration and photosynthesis rates if the SM decreases to 10% [64]. To generate WW and DS conditions, each pool was divided into 400 plots with equal area, and 14 plants of each inbred line were grown in the plot. Maize plants in one pool were well-watered and the soil moisture was kept at around 25%; plants in the other pool were drought-stressed. SM was monitored by using a Delta-T HH2 moisture meter (Delta-T Devices, Ltd). Once the SM of the DS pool reached 10%, the top fully expanded leaves of at least three plants (at around 6-leaf stage) from each plot were pooled and snap-frozen in liquid nitrogen and stored at − 80 °C for later metabolite extraction. The other plants were harvested for fresh weight measurement. The fresh shoots were further maintained at 105 °C for 2 h before being dried at 65 °C. Dry mass was determined until the weight did not lose anymore. The biomass measurements under WW and DS conditions were repeated two times and more than three plants per inbred line in each measurement were used. The average biomasses of the maize population are shown in Additional file 2, Table S1.

### Metabolic profiling based on liquid chromatography-mass spectrometry

Metabolite extraction and measurement of maize leaves were performed with LC-MS method essentially as described by Wu et al. [65]. Then, 20 mg freeze-dried powder of each sample was mixed with 1 ml pre-cooled methanol/methyl-tert-butyl-ether (1: 3), shaked for 15 min at 4 °C. Then the homogeneity was incubated in an ice-cooled ultra-sonication bath and was supplemented with a mixture of 650 μl methanol/water (1: 3). After vortexed and centrifugal for 5 min at 13,523 rcf in 4 °C, the sample was in a phase separation and a 500-μl aliquot from the lower aqueous phase was taken and dried in vacuum. The dried samples were then shipped to Beer-Sheva, Israel, for metabolic profiling. After resuspension with 100 μl 80% MeOH, the samples were analyzed with a

HSS T3 $C_{18}$ reversed-phase column (100 mm × 2.1 mm × 1.8 μm particles; Waters) on a Waters Acquity UPLC system Waters (http://www.waters.com). The mobile phases were 0.1% formic acid in $H_2O$ (A) and 0.1% formic acid in acetonitrile (B). The injection volume is 5 μl. The gradient was as follows: 99:1 V/V at 0 min, 99:1 V/V at 1 min, 60:40 V/V at 11 min, 30:70 V/V at 13 min, 1:99 V/V at 15 min, 1:99 V/V at 16:00, 99:1 V/V at 17 min, 99:1 at 20 min; flow rate, 0.4 ml/min; temperature: 40 °C.

The mass spectra were acquired by a Thermo Fisher Q Exactive (Thermo Fisher, http://www.thermofisher.com). Each extract was separately analyzed with both positive and negative modes of electron spray ionization, covering a mass range from 100 to 1500 m/z. The resolution was set to 70,000. The capillary voltage was set to 3.5 kV with a sheath gas flow of 60 and an auxiliary gas flow of 20 (values are in arbitrary units). The capillary temperature was set to 275 °C, and the drying gas temperature was set to 300 °C. In addition to the full MS scan, another MS scan focused on the top-3 features from each previous scan was performed with the normalized collision energy 25.

Chromatograms from the UPLC-LC-ESI-MS/MS runs were analyzed and processed with Refiner MS 11.0.3 (GeneData, http://www.genedata.com). Molecular masses, retention times, and associated peak intensities for each sample were extracted from .raw files. Chemical noise was subtracted automatically and a pairwise alignment-based tree was applied in the alignment of chromatogram with the *m/z* windows of five points and retention-time windows of five scans within a RT search interval of 0.5 min. Peaks originating from the same molecule different isotope patterns were clustering on the basis of isotope patterns using "chromatogram isotope clustering" activity in Refiner MS (RT tolerance: 0.015 min, m/z tolerance: 5 ppm). Clusters corresponding to the same molecule with different adducts were performed by "adduct detection" activity with mass tolerance as 50 ppm and RT tolerance as 0.1 min, the adduct list in negative ionization mode is $[M-H]^-$ and $[M + Hac-H]^-$ while the list is $[M + H]^+$, $[M + Na]^+$, and $[M + NH_4]^+$ in positive ionization mode. Day normalization and sample-median-normalization were conducted, and the resulting data matrices were used for further analysis.

### Metabolite annotation and identification

The metabolites were putatively annotated based on accurate MS, MS/MS spectra, and the comparison of retention time (RT) with some published papers in maize [30, 43, 66]. To further annotate metabolites, the signals that show similar fragmentation patterns with the reported metabolites were used to query the MS/MS spectral data taken from the literature or to search the databases including mzCloud, MassBank, HMDB, KNApSAcK, and METLIN.

### Statistical analysis of metabolic traits

Metabolites with > 20% missing values under either WW or DS conditions were removed followed by a three-step process which was performed to remove the redundant peaks. At the initial stage, we grouped the metabolite features derived from the same metabolite in either positive or negative ionization mode based on the following criteria: (i) the retention-time tolerance was 0.2 min; (ii) the *m/z* of a smaller peak was identified in MS/MS spectrum in the same time-window of the larger peak; (iii) the

correlation between metabolite features across all inbred lines higher than 0.95. Subsequently, the metabolite datasets from both positive and negative ionization modes were merged by criteria i (retention time) and iii (correlation). Checking the dataset manually and removing the remaining redundant metabolites was the last step.

The signal intensity of each mass feature across the population was log transformed. Principal component analysis (PCA) was performed using the R package "pcaMethods" [67] to evaluate the influence of drought to maize metabolome. Orthogonal partial least squares discriminant analysis (OPLS-DA) was conducted by the R package "ropls" [68] to identify the major discriminant metabolite features between WW and DS conditions. A metabolite feature was considered as a candidate drought-responsive metabolic trait when matching the following criteria: (i) Variable Importance for the Projection (VIP [45];) calculated from OPLS-DA $\geq$ 1; (ii) false discovery rate (FDR) of paired *t*-test in WW and DS conditions $\leq$ 0.05; (iii) |Fold Change (DS/WW)| $\geq$ 2.

### Predict drought indices using metabolic trait

We applied the R package rrBLUP [25] to predict drought indices using the metabolic traits. The survival rate of the maize population was published in 2016 [9], and we selected inbred lines with survival rate > 0 in rrBLUP metabolic prediction. To obtain the best linear unbiased prediction (BLUP) of fresh weight and dry mass under DS, the linear mixed effects function lmer in the lme4 R package was fitted to each genotype: $Y_{ijk}$ = $u$ + Environment$_i$ + Replicate (Enviroment)$_{ij}$ + Genotype$_k$ + e$_{ijk}$. Drought tolerance prediction of three drought indices (survival rate, fresh weight, dry mass) by 1035 drought-responsive metabolic traits (m1), all 3890 metabolic traits (m2), and randomly selected 1035 metabolic traits (m3) were performed independently with rrBLUP. A ten-fold cross-validation was applied and repeated 100 times to compare the performance of the predictions. A bootstrap significance testing approach was applied to estimate the statistical significance of the prediction difference with function *boot.pval()* (theta_null = 1, type = "basic") in R package "boot.pval" [69]. The origin hypothesis is the ratio of two groups of prediction accuracy is equal to 1. Bootstrap replicates were generated with the function *boot()* (R = 100, stype = "w," sim = "ordinary") in R package "boot" [70]. A stepwise regression strategy was applied to select drought-responsive metabolite markers. The prediction variables of maize survival rate were selected from 1035 drought-responsive metabolites under drought conditions. A stepwise regression analysis was conducted with the function stepAIC () in the R package "MASS" (direction = "both," steps = $n$/10, $n$ = sample number) [71]. For comparison of performances, a five-fold cross-validation scheme was promoted and repeated 10 times by using the function *crossval* () in the R package "bootstrap" [72].

### GWAS of metabolic traits

A total of 1,236,497 SNPs [24] with MAF $\geq$ 0.05 of the 385 maize inbred lines were used for GWAS of 1035 drought-responsive metabolites. The mixed linear model in TASSEL 5.0 was used [73, 74]. The Kinship and principal components (PC) were calculated by TASSEL 5.0 with all SNPs used. The first 5 PCs were applied for Q estimating. GWAS was independently performed for each log transformed drought-responsive metabolic trait under both WW and DS conditions. Because of the non-independence

of SNPs caused by strong LD, it is usually too strict for significant association detection when the threshold is derived from the total number of markers [9, 75, 76]. The independent marker number (474,242) was determined by the GEC software [75]. Therefore, we used the suggestive threshold of this association mapping panel ($P < 2.11 \times 10^{-6}$, or 1/474,242) to control type I error rate. Because of the rapid LD decay of the SNP markers in the association panel, candidate genes were identified when the significant SNPs were located within the gene [9]. Genomic regions associated with metabolite traits were identified as previously described [65]. In brief, SNPs associated with drought-responsive metabolites were extracted with a threshold of $P < 2.11 \times 10^{-6}$ before they were assigned to the same group (mQTL) if the physical distance between them was less than 10 kb.

### Network analysis based on metabolome and transcriptome

A gene-metabolite correlation network was constructed using the intersect lines of expression profiles (Dai et al., unpublished) and 1035 drought-responsive metabolic profiles grown under WW and DS conditions. An ordinary least squares (OLS) linear model was used to estimate relationship between each pair of metabolite-gene using the function *lm()* in R with the significant threshold $P \leq 1 \times 10^{-5}$.

Pearson correlation coefficient (PCC) between genes was calculated to construct gene-gene co-expressed networks. *P* value of each gene-gene pair was calculated by the *rcorr()* function in R package "Hmisc." A threshold for building edges between genes in network under WW or DS conditions was $P \leq 4.67 \times 10^{-9}$.

### Detection of hub genes in gene-metabolite correlation/association networks

A permutation test was performed to assess the statistical significance of the deviation of the metabolite number associated with per gene from the expectation based on chance events. In the permutation, each metabolite was assigned randomly to a candidate gene, and the number of signals for each gene was recorded. After 1000-permutation test, the value of significant ($P < 0.01$) number per gene in gene-metabolite correlation network was 7 under WW conditions and 9 under DS conditions, while the number in gene-metabolite association network was 6 under WW conditions and 7 under DS conditions. Therefore, the threshold defining hub genes was the genes correlated with 7 metabolites under WW conditions and 9 metabolites under DS or associated with 6 metabolites under WW conditions and 7 metabolites under DS conditions.

### Enrichment analysis

The gene symbol was transformed to entrez id and KEGG pathway enrichment was performed with web-based toolkit KOBAS [77], and the significance was determined by the Fisher test. The maize metabolic pathways were downloaded from cornCyc 9.0 (https://corncyc-b73-v3.4.maizegdb.org/) [78], and pathway enrichment was performed by an in-house *R* script.

### Identification and functional validation of the CACTA TE in *Bx12*

Primers used for identifying the CACTA TE in *Bx12* in the maize populations have been published before [33]. To investigate the roles of the CACTA TE in *Bx12* in maize drought tolerance, two F2 populations were sowed in the cultivation pools under DS conditions. Leaves were harvested for DNA extraction at V1 stage to genotype the CACTA TE in the populations. Followed by DS, the survival rates of the two populations were recorded after re-watering. Furthermore, we obtained a heterogeneous inbred family (HIF) population based on the CACTA TE at *Bx12* locus from the previously reported recombinant inbred line (RIL) populations derived from Zong3 × Teosinte [34]. After germination, the TE insertion and deletion plants of the HIF population were determined by PCR and transplanted side by side in a cultivation box (30 × 40 × 20 cm, length × width × depth) filled with 7.5 kg of premixed soil (field soil:nutrient soil = 1: 1). Drought treatment was applied to the soil-grown plants at V4 stage by withholding water. Approximately 25 days later, watering was resumed to recover plants showing visible leaf curling and severe stress symptoms, and the number of survived plants was recorded 5 days later.

### Transgenic maize construction and phenotypic analyses

The coding sequence of *ZmGLK44* was amplified from B73 cDNA by Phanta DNA polymerase (Vazyme) with primers ZmGLK44-OE-F/R, and then inserted into pZZ0153-RD101p-*3HA* vector (linearized with enzyme Asc I) [10] by using the recombination kit (Vazyme) to generate the drought inducible expression cassette RD101p-*ZmGLK44-3HA*. The maize transformation was conducted as the previous report [10, 79]. To find out the roles of *ZmGLK44* in maize drought response and drought tolerance, the transgenic plants were grown in the pots containing equal amount of soil (peat:perlite (70:30); Ari Sone Green Inc., Israel) in a walk-in growth room kept at 22–26 °C, with a 16-h light/8-h dark photoperiod under 500 $\mu mol\,m^{-2}\,s^{-1}$ photosynthesis active radiation (ELIXIA LED light system; Heliospectra Inc., USA). The plants were irrigated continuously with water supplemental with nutrient solution (OR 4:2:6 1:500; Deshen GAT Inc., Israel). At 16 days post germination, pretreatment measurements of gas exchange and chlorophyll content were taken and watering was stopped for 15 days until leaf curling and severe stress symptoms were visible for all plants. Then, irrigation was renewed for 6 days. Following the recovery treatment, the plants were analyzed for gas exchange, survival, ion leakage, and chlorophyll content. Gas exchange was measured using the LI-6800 portable gas exchange system (LI-COR Inc., USA) on fully expanded leaves at saturating light (1200 $\mu mol\,m^{-2}\,s^{-1}$) with 400 $\mu mol\,mol^{-1}$ $CO_2$ surrounding the leaf (Ca) and 10% blue light of photosynthetically active photon flux density. The flow rate was set to 500 $\mu mol\,air\,s^{-1}$. Leaf temperature was 26 °C. Chlorophyll content was measured using handheld chlorophyll meters (CCM-200 (Opti-Science Inc., USA) system). Ion leakage was measured by sampling whole shoots in tubes filled with 40 ml distilled DI water, followed by an overnight gently shaking incubation at room temperature and quantification of conductivity using conductivity meter CD-4301 (Lutron Inc., Taiwan, China) before and after 60 min boiling of the samples at 100 °C.

## Quantification of targeted metabolites for transgenic plants based on HPLC-ESI-Q TRAP-MS/MS

Metabolites correlated with the expression of *ZmGLK44* in AMP population and participated in L-tryptophan biosynthesis pathway under well-watered and drought conditions were quantified with an HPLC-ESI-Q TRPA-MS/MS system (HPLC; Shim-pack UFLC SHIMADZU CBM20A system, www.shimadzu.com.cn; MS, Applied Biosystems 4000 Q TRAP, www.appliedbiosystems.com.cn). Instrumental settings were described previously [80]. The analytical conditions were as follows. HPLC: solvent system, water (0.04% acetic acid): acetonitrile (0.04% acetic acid); gradient program, 95:5 V/V at 0 min, 5:95 V/V at 15.0 min, 5:95 V/V at 17.0 min, 95:5 V/V at 17.1 min, 95:5 V/V at 20.0 min; column, shim-pack VP-ODS C18 (pore size 5.0 μm, length 2 × 150 mm). Injection volume, 5 μl; flow rate, 0.30 ml/min; temperature, 40 °C. ESI-Q TRAP-MS/MS: Linear ion trap (LIT) and triple quadrupole (QQQ) scans were acquired on a triple quadrupole-linear ion trap mass spectrometer (Q TRAP), ABI 4000 Q TRAP LC/MS/MS System, equipped with an ESI Turbo Ion-Spray interface, operating in both positive and negative ion mode. The ESI source operation parameters were as follows: ion source, turbo Spray; source temperature 500 °C; ion spray voltage (IS) 5500 V; ion source gas I (GSI), gas II (GSII), and curtain gas (CUR) were set at 55 psi, 60 psi, and 25.0 psi; the collision gas (CAD) was high. The MRM transition of each metabolite is listed in Additional file 3, Table S18. Data were processed using Analyst 1.5 software; peak areas were integrated by the IntelliQuan algorithm. The drought treatment and metabolite extraction were described as before.

## Exogenous L-tryptophan and DIMBOA treatment

Exogenous L-tryptophan treatment was performed with 10-day-old maize B73 and B104 seedlings. Plants were grown in 0.1-L pots containing equal amount of soil (peat: nutrient soil: sand = 1:1:1). L-tryptophan (Coolaber, Beijing) were dissolved in 1% (m/v) KOH solution and diluted to 50 μg/ml with water. Five days before drought treatment, 200 ml of tryptophan solution with 0.1% Tween-20 was sprayed onto the leaves of the maize plants (*n* = 72) every day and the control plants were sprayed with 0.1% Tween-20 solution. Then the plants ceased to be watered for about 10 days until leaf curling and severe stress symptoms were visible. Irrigation was renewed to recover the plant and the survived plants were counted 3 days later. Six plants of each line were compared in each test, and statistical significance was determined based on data from six independent experiments.

Exogenous DIMBOA treatment was performed with 10-day-old maize B73 seedlings grown under the same condition as plants in L-tryptophan treatment. DIMBOA (Rechemscience, Shanghai) were dissolved in DMSO to 10 mg/ml and diluted to 20 μg/ml with water. In total, 200 ml of DMBOA solution with 0.1% Tween-20 was sprayed onto maize leaves (*n* = 72) for 5 days and the controlled plants were sprayed with solution contained the same amount of DMSO and Tween-20. Drought treatment, survival rate investigation, and statistical analysis process of these plants were the same with plants treated with L-tryptophan. For plants grown under well-watered conditions, 40 DIMBOA-treated and control plants were marked and their height was measured at 5 and 10 days after DIMBOA feeding. Top expand leaves of three plants were harvested

and mixed with $H_2O_2$ and MDA measurement as previously described [79]. Shoot of four plants were harvest, weighted (fresh weighted), and dried at 65 °C for constant weight to obtain dry mass.

### RNA-seq and RT-qPCR analysis of *ZmGLK44* transgenic lines

Positive transgenic plants and its negative siblings were planted alongside in the cultivation boxes under WW and DS conditions. The top fully expanded leaves were harvested when the SM in the DS boxes reached ~ 10%. Total RNAs were extracted by using the Trizol kit [81]. RNA-seq was adopted at BGI (Shenzhen, China) via a 150-bp paired end sequencing method. An average of ~ 14 million reads were mapped to the maize reference transcriptome v3.31 and the gene expression levels were evaluated by using RSEM [82]. Differential expressed analysis was performed using the R package "*DESeq2*" [83].

For semi-quantitative reverse transcription PCR (qRT-PCR) analyses, MLV reverse transcriptase (Promega) was used to synthesize the first-strand cDNA. Real-time PCR was conducted with SYBR Green Mix (Vazyme) in a CFX96 Real-Time System (Bio-Rad), and maize actin gene was used as an internal control. Expression of *ZmGLK44* in transgenic plants were determined by primers ZmGLK44-F and HA-R. Genes involved in ABA signaling pathway including *ZmPYL12*, *ZmSnRK2.1*, *ZmSnRK2.4*, *ZmPP2C-A4*, *ZmPP2C-A7*, *ZmRD17*, *ZmRAB18*, and *ZmLEA2* were chosen for expression validation with primers Actin-qPCR-F/R, ZmPYL12-qPCR-F/R, ZmSnRK2.1-qPCR-F/R, ZmSnRK2.4-qPCR-F/R, ZmPP2C-A4-qPCR-F/R, ZmPP2C-A7-qPCR-F/R, ZmRD17-qPCR-F/R, ZmRAB18-qPCR-F/R, and ZmLEA2-qPCR-F2/R2 in RT-qPCR analyses.

### Luciferase activity assay in maize protoplasts

The coding sequence of *ZmGLK44* was amplified by RT-PCR with primers pRTL2-GLK44-F/R and inserted into pRTL2 vector (linearlized with enzyme BamH I) by using the recombination kit (Vazyme) to generate pRTL2-*ZmGLK44*, while promoter fragments of *PAT1*, *ASA2*, and *TSB2* were amplified from B73 genome with primers ASA2p-LUC-F/R, PAT1p-LUC-F/R, TSB2p1-LUC-F/R, TSB2p2-LUC-F/R, and then inserted into pGreenII0800-LUC vector (digested with enzyme Xho I) by using the recombination kit (Vazyme) to generate pGreenII0800-*PAT1-LUC*, pGreenII0800-*ASA2p-LUC*, pGreenII0800-*TSB2p1-LUC*, and pGreenII0800-*TSB2p2-LUC*. Maize protoplast isolation and transformation were performed according to the protocol developed by Jen Sheen lab (http://genetics.mgh.harvard.edu/sheenweb/). After 16 h of incubation at 22 °C, luciferase activities were measured by a Luciferase reporter system (Promega) in SpectraMax i3x (Molecular Devices).

### Primers

All the primers used in this study are listed in Additional file 4, Table S19.

### Supplementary Information

---

**Additional file 1: Figures S1-S14.**

**Additional file 2.** A master file for Table S1-13, about the information of the association population, the metabolomes, mGWAS, candidate genes and their genetic control.

---

**Additional file 3** A master file for Table S14-18, about the information of two validated genes *Bx12* and *ZmGLK44*, and stepwise regression with m-traits and survival rate under drought stress.

**Additional file 4.** Oligos used in this study. Table S19.

**Additional file 5.** Review history.

#### Acknowledgements
We thank doctors Haiyang Wang (South China Agricultural University), Yi Wang (China Agricultural University), Weiwei Wen, Yingjie Xiao (Huazhong Agricultural University), and Yalong Guo (Institute of Botany, the Chinese Academy of Sciences) for critical reading and suggestions for the manuscript, Feng Tian (China Agricultural University) for providing DNA of landraces and Miss Lili Qi for culturing some plants.

#### Review history
The review history is available as Additional file 5.

#### Peer review information

#### Authors' contributions
M.D., Y.B., and W.C. designed the research. F.Z. and J.W. performed the experiments. N.S., S.W., A.E., A.F., Z.Z., J.Y., and F.Q. also performed experiments or analyzed the data. M.D., W.C., Y.B., F. Z., and J.W. analyzed the data and wrote the manuscript with significant input from all other authors. The authors read and approved the final manuscript.

#### Funding
This work was supported by grants from Wuhan Applied Foundational Frontier Project (2020020601012258), the National Natural Science Foundation of China (32061143031), Beijing Outstanding Young Scientist Program (BJJWZYJH01201910019026), the Fundamental Research Funds for the Central Universities (2662020SKY009), the 111 Project Crop genomics and Molecular Breeding (B20051), and the Baichuan Project at the College of Life Science and Technology, Huazhong Agricultural University.

#### Availability of data and materials
The RNA-seq reads generated in this study have been deposited to the Genome Sequence Archive in BIG Data Center, Beijing Institute of Genomics (BIG), Chinese Academy of Science, and are publicly accessible at https://ngdc.cncb.ac.cn/gsa under accession number CRA004467 [84]. The metabolomic raw data reported in this paper have been deposited in the OMIX, China National Center for Bioinformation/Beijing Institute of Genomics, Chinese Academy of Sciences, and are accessible at https://ngdc.cncb.ac.cn/omix under accession number OMIX419 [85]. The variant map of the Maize Association Panel was previous published [86] and the genotyping set (with hapmap format) are accessible at http://maizego.org/Resources.html.

## Declarations

#### Ethics approval and consent to participate
Ethics approval is not applicable.

#### Competing interests
The authors declare no competing financial interests.

#### Author details
[1]National Key Laboratory of Crop Genetic Improvement, Huazhong Agricultural University, Wuhan 430070, China. [2]Hubei Hongshan laboratory, Wuhan 430070, China. [3]School of Plant Sciences and Food Security, The Institute for Cereal Crops Improvement, Tel-Aviv University, 69978 Tel Aviv, Israel. [4]Department of Genetics, Stanford University School of Medicine, Stanford, CA 94305, USA. [5]Max Planck Institute of Molecular Plant Physiology, 14476 Potsdam, Germany. [6]State Key Laboratory of Plant Physiology and Biochemistry, College of Biological Sciences, China Agricultural University, Beijing 100193, China. [7]Department of Life Sciences, Ben-Gurion University of the Negev, 8410501 Beersheba, Israel.

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

## 