## [**Additional file 5.** Review history. · Genome Biology]

Review History

First round of review

Reviewer 1

Are you able to assess all statistics in the manuscript, including the appropriateness of statistical tests used? No, I do not feel adequately qualified to assess the statistics.

Comments to author:

This manuscript describes a large-scale metabolomic analysis of a genotyped population of maize inbred lines, with and without drought stress. The results will provide an important resource for further research on metabolomic changes induced in maize by drought. Although this is useful information, there are some improvements that need to be made to the manuscript:

- A. The methods are not described adequately for others to repeat them. I mention a few examples below, but the authors should consider the methods section very carefully to determine whether a plant molecular biologist could repeat these experiments with the information that is provided.
- B. Large data sets that underlie important conclusions in this manuscript are available "upon request". They should be placed in public databases.
- C. Two genes, Bx12 and ZmGLK44, were examined in more detail. However, as described below, there are some problems with the methodology and conclusions.

Specific comments:

1. It would be helpful if the authors wrote out the name or the described the function of key genes that play a central role in their findings. For instance, ZmCS1 and ZmCS2 are never described as encoding citrate synthase, except in Figure S3. This is confusing because ZmCS1 can also mean chorismite synthase, e.g. Babu et al, *Plant Breeding* 131, 20—27 (2012)
2. GRMZM2G063909 and GRMZM2G064023 (CS1 and CS2) are described as citrate synthases in this study. Arabidopsis AT2G44350 is described as the closest Arabidopsis ortholog (Table S9). Is the converse also true, i.e. are GRMZM2G063909 and GRMZM2G064023 the most similar maize genes to AT2G44350, or are there other possible citrate synthases encoded in the maize genome? This is important if the authors are assigning citrate synthase as a function for these two genes. As far as I know, the actual enzymatic function has not been confirmed for GRMZM2G063909 and GRMZM2G064023 and previous research indicates that these two enzymes may have other activity.
3. Line 315-317: "Previous reports have shown that an InDel of CACTA TE (TE_InDel) in Bx12 is the causal variant that controls the divergence of Bx12 expression and further DIMBOA-Glc production in maize leaves [35]," This CACTA-family transposon insertion in the Bx12 gene was originally reported by Meihls et al, 2013 (reference #78). It would be appropriate to cite this publication, rather than the current reference #35, which discovered the same transposon insertion six years later.
4. Line 335-337: "Spraying DIMBOA to the maize genotype B73 decreased the survival rates of

maize plants after drought treatment, indicating negative roles of DIMBOA and DIMBOA-Glc in maize drought tolerance" This experiment does not show that DIMBOA has any direct relevance to drought tolerance. It only shows that DIMBOA has a negative effect on maize plants. This is not surprising because DIMBOA causes oxidative stress and also induces secondary defense responses. At a minimum, a control experiment showing that DIMBOA sprayed on non-drought-stressed plants is necessary. However, my prediction is that there will be negative effects on non-drought-stressed maize plants if they are sprayed with DIMBOA.

5. Line 337-339: "Together, these data suggested that Bx12 played positive roles in maize drought tolerance likely through down-regulation of DIMBOA-Glc accumulation upon drought stress." The authors assume that lower abundance of DIMBOA-Glc provides protection against drought stress. The opposite hypothesis, that higher abundance of HDMBOA-Glc provides protection against drought stress also needs to be considered. HDMBOA-Glc is produced from DIMBOA-Glc by BX12.

6. Lines 374-375 "Based on our data sets, ZmGLK44 was a hub gene correlated / associated with 30 drought-responsive metabolites (Fig. 5a; Additional File 13, Table S9)." Figure 5a shows only 27 metabolites. What are the other three metabolites?

7. Line 517-519: "More than 60% of the phenotypic variance of survival rate could be predicted by combined 15 m-traits (Additional File 25, Figure S10; Additional File 26, Table S16)," This is a result. Results should be brought up first in the results section. The discussion should be limited to discussing the significance of the results, rather than presenting new results.

8. The authors only discuss measurements of DIMBOA-Glc was HDMBOA-Glc detected in these assays? For instance, Figure S6 should include analysis of HDMBOA-Glc.

9. Figure 4a is misleading because it suggests that only HDMBOA-Glc is broken down by a glucosidase. There would be similar glucosidase-mediated breakdown of DIMBOA-Glc, DIM2BOA-Glc, and HDM2BOA-Glc.

10. Table S15 lists about 1,500 genes that are regulated by ZmGLK44, but there is no way to determine which of these genes the authors placed in the different categories that are listed as overexpressed in Figure 5c. Without knowing this, it is not possible to determine the overall value of the gene categories listed in Figure 5c. Perhaps the genes in Table S15 could be marked to show which category they belong to in Figure 5c.

11. Table S15: At the bottom of this table there are some genes with an expression level of zero with or without ZmGLK44 expression. How can these be considered differentially expressed?

12. Figure 5c: Glycerophosphodiester degradation is listed as an enriched category. What genes are actually in this category? Is it possible that this is also tryptophan biosynthesis? A key step in tryptophan biosynthesis is indole-3-glycerolphosphate degradation.

13. Figure 5c: Both tryptophan biosynthesis and benzoxazinoid biosynthesis are differentially regulated by ZmGLK44. However, based on other information provided by the authors, this

could have opposite effects on drought tolerance. Tryptophan increases drought tolerance and benzoxazinoids (DIMBOA-Glc a downstream product of DIBOA-Glc, a low-abundance intermediate) decreases drought tolerance. It would be good to consider more carefully which benzoxazinoid biosynthesis genes are induced by ZmGLK44 and not just focus on tryptophan biosynthesis. For instance, is benzoxazinoid biosynthesis up-regulated or down regulated by ZmGLK44? What about Bx12 specifically?

14. Figure 5d: It would be good to also connect this pathway to DIBOA-Glc biosynthesis, which is listed as enriched in Figure 5c. Indole is a branch point for the biosynthesis of tryptophan and benzoxazinoids. See for instance: Richter et al, Plant Journal, 2021, 106:245-257. doi: 10.1111/tpj.15163

15. Line 403-404: "(h) Activation of TRP2p1::LUC, but not the other promoter fragments, by ZmGLK44 in maize protoplasts." In the figure TRP2p2::LUC is activated (not TRP2p1::LUC)

16. Figure 5e: There are at least two tryptophan synthase beta subunit (TRP2) genes in maize. Which one is shown in this figure?

17. Figure 5e,h: There are at least three genes that encode tryptophan synthase alpha subunit activity (indole synthase; TRP1 in the figure) in maize. The authors should specify which gene they are looking at here. Ideally, the authors should look at expression of all three indole synthase genes (TSA, BX1, and IGL).

18. Figure 5d. TRP1 and TRP2 (tryptophan synthase alpha and beta subunits) are more commonly called TSA and TSB in maize. See for instance: Kriechbaumer et al, BMC Plant Biology, 2008, 8:44. doi: 10.1186/1471-2229-8-44.

19. Line 494,499 lepidopteran should not be italicized

20. Line 498-499: "breeding maize cultivars with both drought tolerance and insect (e.g. lepidopteran species) resistance." This would likely cause sensitivity to other insects (aphids).

21. Lines 687-680: "LC-MS data were obtained using a Waters Acquity UPLC 679 system (Waters, <http://www.waters.com>), coupled to a Thermo Fisher Q Extractive (Thermo Fisher, <http://www.thermofisher.com>)." This is not enough information. The authors need to specify what column was used, what gradient, what mass ranges were measured, etc. I should be able to repeat this experiment based on what is written in the methods section.

22. Lines 686-687: "The further processing of the MS data included isotope clustering, adduct detection, and library search." Again, not nearly enough information. I would not be able to repeat this analysis based on the information that is provided.

23. Methods section: Although there is a description of the exogenous tryptophan treatment, there is no description of the method that was used for exogenous DIMBOA treatment. Given the reactive nature of DIMBOA and its spontaneous degradation to other metabolites, it should also be noted where DIMBOA was purchased and what purity it had at the time that it was added to

maize plants. The half-life of DIMBOA in an aqueous solution has been reported to be about 24 hours.

24. Table S3 lists only the 1035 metabolites that the authors deem to be drought-responsive. For the scientific record, negative evidence is also important. Also, some readers may have an interest in a particular mass feature that the authors did not conclude to be drought-responsive. Please include all 3,890 reliably detected mass features in this table.

25. Table S5. What about the mQTL for the metabolites that the authors did not deem to be drought-responsive? There is a lot of information that would results from a more complete analysis of the metabolomic data set that is just being ignored. The 1035 chosen metabolites are just based on some arbitrary cutoff that the authors have determined. Readers will also be interested in the mQTL affecting the other 2,855 metabolites that were detected in these assays. Please include this information in Table S4.

26. Table S14. A putative metabolite is described as "34N" (in cell C10 of the spreadsheet). That is not a very informative metabolite description.

27. Table S14: PP_Group_00384 is described as likely being tryptophan (line 7). However, in line 41 a different metabolites is described as being tryptophan.

28. Line 879-880: "All other reasonable requests for data and research materials are available by contacting the corresponding authors." All raw data for large data sets, e.g. RNAseq and metabolomics should be placed in a public database. RNAseq data should be in GenBank short read archive. Metabolomic raw data (.raw files) should be placed in Dryad, Cyverse or some other public database. If ten or twenty years from now someone wants to look at these data, there is no certainty that the authors will still have the data and will be able to respond to a "reasonable request."

29. Is the large SNP dataset for the inbred line population available from a public database? If so, which database? If not, it should be made available.

30. The English-language writing is generally good, but there are occasional lapses. I recommend that one of the native English-speaking co-authors should do some careful proofreading. For instance, a sentence like "Of the upregulated metabolites, there were tryptophan (Trp) and its relative metabolite Achillamide derivatives." is not quite grammatically correct.

Reviewer 2

Are you able to assess all statistics in the manuscript, including the appropriateness of statistical tests used? Yes, and I have assessed the statistics in my report.

Comments to author:

Genomic basis underlying the metabolome-mediated drought adaptation of maize

The Abstract should include which tissues or organs of the maize plant were profiled for metabolite and transcript abundances.

Line 110, change from principle to principal component analysis

Lines 136-139, this is a far too sweeping statement especially given that this is an artificial "drought" study in pots. It needs to be couched as upregulated for this environment. Let's see the results across multiple field environments before concluding as such. Also, there is way to conclude that these shifts in metabolites are causal in conferring drought tolerance. They are responding. That is all that can be concluded.

Lines 151-154, these rrBLUP results with metabolites would be better served with the inclusion of a comparison to prediction of the dry mass, fresh weight, and survival rate phenotypes with genome-wide genetic markers. This population has been scored with SNP markers, so it is a straightforward comparison. The focus on significance here is misleading, as the focus should be on prediction ability/accuracy.

How is this statistical test carried out anyway? It should not be a t-test. The rationale for not using a t-test is that the magnitude of the denominator of the t-statistic is dependent on the number of resampling runs (standard error of the difference in means). Therefore, if performing a sufficient number of resampling runs, the denominator would be very small, and it could then be possible to obtain a significant p-value for any comparison with an arbitrarily small difference. Another way of understanding the problem with the t-test is that because each fold is drawn from the same original dataset, the different folds are not independent of each other, which violates the t-test assumption that observations are independent. I suggest using a bootstrap significance testing approach, which I believe to be valid in this case. If properly conducted, it should be independent of the number of resampling runs.

Line 162, kinship not kinships

Line 164, need to report what multiple test correction approach was used. If none was used, then either use Bonferroni or FDR. Same for eQTL mapping. The 1/X method is too permissive.

Lines 175-176, concluding many new putative drought-responsive QTL is a bit of a stretch. By no means was the maize drought linkage/GWAS mapping literature deeply surveyed.

Line 178, need to define "lead SNP". I assume this is a peak SNP - a SNP with smallest P-value at a locus, but it still needs to be defined.

Lines 179-180, a statistical test of enrichment needs to be conducted to say anything meaningful about finding 190 TFs.

Fig 2a - nadir view DS Manhattan plot totally unnecessary and less than helpful. Flip upright.

Line 192, I have never heard of it referred to as a permutation assay. Permutation test?

Lines 212-217, I do not agree with the candidate gene only analysis in this context. It is more important to see how these associated genes rank genome-wide rather than compared to a random subset. I am not convinced on reliability of results as stated in lines 217-219 based on the collectively results. It is easy to subsample data in different ways with biological hypotheses, but the presentation and argument thereof needs to be better conveyed and supported beyond correlative machinery. Otherwise, it is a weakened argument.

Line 258, 1Mb window really needs to be used for cis-eQTL and is more often the standard than 20kb despite the citation. This is resulting in inflated number of trans-eQTL.

Lines 272-273, static is misleading, as it implies that it is constitutive. Shared? Anyway, something else is needed.

Lines 285-286, "expression regulation of themselves..." not clear. Please revise for clarity.

Point-by-point responses to reviewers

Editor's comments

As you will see from the reports, both referees are broadly favorable and find the work of potential interest, but they raise important issues that we must ask you to address, in the form of a revised manuscript, before we reach a final decision on publication. In particular, it seems to us to be essential that both Referees' concerns regarding the methods and analyses resolved. Referee 2 has also requested additional statistical analyses, and Referee 1 also has raised concerns about the experiments with *Bx12* and *ZmGLK44*. Please ensure that these and all other issues raised by the referees are addressed in full. Please make sure all data is deposited in a public repository and at least made available for reviewers on re-review.

Reply: We really appreciate the insightful comments and valuable suggestions from the editor and the reviewers. We added more information of methods and analyses as suggested by both reviewers. We conducted more experiments or carefully addressed the concerns or suggestions about *Bx12* and *ZmGLK44* raised by reviewer #1. We also performed more statistical analyses as suggested by reviewer #2. As regarding the source data, we deposited them in the public repository, which can be reached easily. Please see our point-by-point responses to the reviewers below.

Reviewer reports:

Reviewer #1: This manuscript describes a large-scale metabolomic analysis of a genotyped population of maize inbred lines, with and without drought stress. The results will provide an important resource for further research on metabolomic changes induced in maize by drought. Although this is useful information, there are some improvements that need to be made to the manuscript:

A. The methods are not described adequately for others to repeat them. I mention a few examples below, but the authors should consider the methods section very carefully to determine whether a plant molecular biologist could repeat these experiments with the information that is provided.

B. Large data sets that underlie important conclusions in this manuscript are available "upon request". They should be placed in public databases.

C. Two genes, *Bx12* and *ZmGLK44*, were examined in more detail. However, as described below, there are some problems with the methodology and conclusions.

Reply: We really appreciate your careful reading, evaluation, and positive comments

on our manuscript. We have tried our best to revise the manuscript according to your insightful comments and invaluable suggestions, and hopefully addressed all your concerns in our revised manuscript. Please see our point-by-point answers below.

Specific comments:

1. It would be helpful if the authors wrote out the name or the described the function of key genes that play a central role in their findings. For instance, *ZmCS1* and *ZmCS2* are never described as encoding citrate synthase, except in Figure S3. This is confusing because *ZmCS1* can also mean chorismite synthase, e.g. Babu et al, Plant Breeding 131, 20-27 (2012).

Reply: Thanks for your suggestions, which are very helpful. Yes, we should give more details for the key functional genes in our previous manuscript. In the main manuscript, we mentioned several key functional genes, including *ZmSKD1*, *ZmCS1*, *ZmCS2*, *Bx12*, *ZmGLK44* etc. We have added the information of these key genes to the revised manuscript, please see line 209: “...*ZmSKD1* (*suppressor of K⁺ transport growth defect 1*)...”, lines 239-240: “...*ZmCS1* (*citrate synthase 1*, GRMZM2G063909), *ZmCS2* (*citrate synthase 2*, GRMZM2G064023)...”, lines 260-262: “...*Bx12* (GRMZM2G023325, encoding an enzyme for the benzoxazinoid metabolic pathway) and *ZmGLK44* (GRMZM2G124540, encoding Goden 2-like TF)...”. The annotation of other functional genes can be found in Table S5, Table S8 and Table S10.

2. GRMZM2G063909 and GRMZM2G064023 (CS1 and CS2) are described as citrate synthases in this study. Arabidopsis AT2G44350 is described as the closest Arabidopsis ortholog (Table S9). Is the converse also true, i.e. are GRMZM2G063909 and GRMZM2G064023 the most similar maize genes to AT2G44350, or are there other possible citrate synthases encoded in the maize genome? This is important if the authors are assigning citrate synthase as a function for these two genes. As far as I know, the actual enzymatic function has not been confirmed for GRMZM2G063909 and GRMZM2G064023 and previous research indicates that these two enzymes may have other activity.

Reply: Thanks for your valuable suggestions, which are helpful to clarify the roles of *ZmCS1* and *ZmCS2* in citrate metabolism. Based on your suggestions, we performed phylogenetic assays for the putative citrate synthases in maize and Arabidopsis. As shown in Figure S5, the putative citrate synthases could be divided into two subclasses, and AT2G44350 (*AtCS4*) belongs to subclass 1 and has three orthologs in maize,

including GRMZM2G063909 (*ZmCS1*), GRMZM2G064023 (*ZmCS2*) and GRMZM2G063851. We reworded our statement in the revised manuscript in lines 239-242 as follows:

“Three hub genes *ZmCS1* (*citrate synthase 1*, GRMZM2G063909), *ZmCS2* (*citrate synthase 2*, GRMZM2G064023), which are two putative citrate synthase genes (Additional File 1, Figure S6), and *ZmMYBR88* (GRMZM2G064328) were statically detected under both WW and DS conditions.”

Figure S6. Phylogenetic tree of putative citrate synthases from maize and Arabidopsis. The gene accession numbers are AT3G58740 (*AtCS1*), AT3G58750 (*AtCS2*), AT2G42790 (*AtCS3*), AT2G44350 (*AtCS4*), AT3G60100 (*AtCS5*) for Arabidopsis, and GRMZM2G063909 (*ZmCS1*), GRMZM2G063851 (*ZmCS2*), GRMZM2G064023, GRMZM2G135588 for maize.

3. Line 315-317: "Previous reports have shown that an InDel of CACTA TE (TE_InDel) in Bx12 is the causal variant that controls the divergence of Bx12 expression and further DIMBOA-Glc production in maize leaves [35], "This CACTA-family transposon insertion in the Bx12 gene was originally reported by Meihls et al, 2013 (reference #78). It would be appropriate to cite this publication, rather than the current reference #35, which discovered the same transposon insertion six years later.

Reply: Thanks for pointing out this. Yes, the original report of the CACTA TE (TE_InDel) in *Bx12* (*Bx10C*) that controls the divergence of *Bx12* expression and further DIMBOA-Glc was from Meihls et al, 2013 (now it is reference #33). We thus cited this reference in the revised manuscript as suggested (line 335).

4. Line 335-337: "Spraying DIMBOA to the maize genotype B73 decreased the survival rates of maize plants after drought treatment, indicating negative roles of DIMBOA and DIMBOA-Glc in maize drought tolerance". This experiment does not show that DIMBOA has any direct relevance to drought tolerance. It only shows that DIMBOA has a negative effect on maize plants. This is not surprising because DIMBOA causes oxidative stress and also induces secondary defense responses. At a minimum, a control experiment showing that DIMBOA sprayed on non-drought-stressed plants is necessary. However, my prediction is that there will be negative effects on non-drought-stressed maize plants if they are sprayed with DIMBOA.

Reply: Thanks for this valuable suggestion. We should show the effects of DIMBOA on plant growth under non-drought conditions in our previous manuscript. As suggested by this reviewer, we did foliar spray of DIMBOA (1% Tween-20 as control) to the maize seedling at three-leaf stage grown under normal conditions to know how DIMBOA affects the growth of maize plants. The method used in this assay was added into the revised manuscript in lines 927-938 as follows:

“Exogenous DIMBOA treatment was performed with 10-day-old maize B73 seedlings grown under the same condition as plants in L-tryptophan treatment. DIMBOA (Rechemscience, Shanghai) were dissolved in DMSO to 10mg/ml and diluted to 20ug/ml with water. 200ml of DIMBOA solution with 0.1% Tween-20 was sprayed onto maize leaves (n = 72) for five days and the controlled plants were sprayed with solution contained the same amount of DMSO and Tween-20. Drought treatment, survival rate investigation and statistical analysis process of these plants were the same with plants treated with L-tryptophan. For plants grown under well-watered conditions, 40 DIMBOA-treated and control plants were marked and their height were measured at 5 and 10 days after DIMBOA feeding. Top expand leaves of three plants were harvested and mixed with H₂O₂ and MDA measurement as previously described [80]. Shoot of four plants were harvest, weighted (fresh weighted), dried at 65°C for constant weight to obtain dry mass.”

Next, we analyzed the plant heights, fresh mass, dry mass, H₂O₂ contents and malondialdehyde (MDA, indicator of oxidation stress) levels of the maize plants grown under normal conditions. The results showed that all these indices did not show significant difference between maize plants treated with or without DIMBOA (Fig S9).

Thus, DIMBOA, at the concentration used here, might have no significant impact on maize growth under normal growth conditions. However, spray of DIMBOA caused sensitivity of maize plants to drought (Fig 4n and o), indicating a negative impact of DIMBOA on maize drought tolerance.

Fig S9. Effects of DIMBOA on maize growth.

(a) Plants treated with or without DIMBOA (Mock treatment). (b) Statistical analyses of the plant height, fresh weight, dry mass, H₂O₂ content and malondialdehyde (MDA) content for the maize plants treated with or without DIMBOA treatment.

Fig 4 (n) and (o) The growth (n) and the survival rate (o) of maize B73 plants after drought-stressed (DS) with or without (CK) DIMBOA treatment. Error bars, s.d. based on 6 biological repeats (n=24); statistical significance was determined by paired *t* test.

5. Line 337-339: "Together, these data suggested that Bx12 played positive roles in maize drought tolerance likely through down-regulation of DIMBOA-Glc accumulation upon drought stress." The authors assume that lower abundance of DIMBOA-Glc provides protection against drought stress. The opposite hypothesis, that higher abundance of HDMBOA-Glc provides protection against drought stress also needs to be considered. HDMBOA-Glc is produced from DIMBOA-Glc by BX12.

Reply: Thanks for pointing out this. We were focusing on DIMBOA-Glc and *Bx12* because DIMBOA-Glc was a drought responsive metabolite in the maize population based on our standard shown in line135 “(|Fold Change| ≥ 2, false discovery rate [FDR] < 0.05)”, while HDMBOA-Glc didn’t show significant difference in the maize population treated with or without drought based on our standard (Reply Figure 1). More importantly, *Bx12* was induced by drought and showed significant association with DIMBOA-Glc only under drought conditions, while it was significantly associated with HDMBOA-Glc under both drought and well-watered conditions, indicating more specific roles of DIMBOA-Glc/*Bx12* pair in maize drought responses (Figure S7). Our analyses might indicate the functional differentiation of DIMBOA-Glc and HDMBOA-Glc, which was consistent with the previous studies showing that DIMBOA and DIMBOA-Glc treatments induced significant callose deposition in the leaves of wheat plants, while HDMBOA-Glc had no effect (Li et al., 2018). We agree with this reviewer that investigating how HDMBOA-Glc functions in environmental stress responses is an interesting topic, but the challenge is the chemical synthesis of HDMBOA-Glc, because it is not commercially available.

Reply Figure 1. HDMBOA-Glc levels in the maize population under drought-stressed (DS) or well-watered (WW) conditions.

Fig S8. Associations of *Bx11* and *Bx12* loci with DIMBOA-Glc levels.

(a)-(c) Manhattan plot showing the associations of SNPs in region Chr1 66-67 Mb with metabolites HBOA (a), HDMBOA-Glc (b) and DIMBOA-Glc (c). (c) and (d) Associations of SNPs in *Bx11* (c) and *Bx12* (d) loci with metabolite DIMBOA-Glc. (e) and (f) Gene models of *Bx11* (e) and *Bx12* (f). (g) LDs among the significant SNPs of *Bx11* and *Bx12* loci. The significant SNPs in *Bx11* and *Bx12* loci are belonging to the same LD block ($R^2 > 0.2$). On average, SNPs in *Bx12* locus show higher LD to each other than those SNPs in *Bx11* locus.

Reference

Li B, Förster C, Robert CAM, Züst T, Hu L, Machado RAR, Berset JD, Handrick V, Knauer T, Hensel G, Chen W, Kumlehn J, Yang P, Keller B, Gershenzon J, Jander

G, Köllner TG, Erb M. **Convergent evolution of a metabolic switch between aphid and caterpillar resistance in cereals.** *Sci Adv.* 2018.4(12):eaat6797.

6. Lines 374-375 "Based on our data sets, *ZmGLK44* was a hub gene correlated / associated with 30 drought-responsive metabolites (Fig. 5a; Additional File 13, Table S9)." Figure 5a shows only 27 metabolites. What are the other three metabolites?

Reply: Thank you very much for your careful reading and checking on this. Yes, *ZmGLK44* was a hub gene correlated / associated with 30 drought-responsive metabolites. We are sorry for missing those three metabolites, which are PP_Group_01992, PP_Group_18074 and PP_Group_02975. Now they were added into Figure 5a in the revised manuscript.

Fig 5(a) Metabolites correlated/associated with *ZmGLK44*. The size of circles represents fold changes of metabolite levels in drought response. Metabolites annotated as amino acid derivatives were indicated as purple circles. 1a: 1-Acetyl-3-carboxy-3,4-dihydro- β -carboline; 1b: ethyl 6-acetamido-4-hydroxyquinoline-3-carboxylate.

7. Line 517-519: "More than 60% of the phenotypic variance of survival rate could be predicted by combined 15 m-traits (Additional File 25, Figure S10; Additional File 26, Table S16)," This is a result. Results should be brought up first in the results section. The discussion should be limited to discussing the significance of the results, rather than presenting new results.

Reply: Thanks for the suggestion. As suggested, we move the Figure S2 (previous

Figure S10) and Table S4 (previous Table S16) to the result section. Please see revised manuscript in lines 155-157 as follows:

“In addition, we found that a combination of 15 m-traits could explain more than 60% of the phenotypic variance of survival rate (Additional File 1, Figure S2; Additional File 2, Table S4).”

8. The authors only discuss measurements of DIMBOA-Glc. Was HDMBOA-Glc detected in these assays? For instance, Figure S6 should include analysis of HDMBOA-Glc.

Reply: Yes, we also detected HDMBOA-Glc in our study, but it was not a differentially-accumulated metabolite (or drought-responsive metabolite) in the maize population based on our standard as shown in lines 132-135: (i) Variable Importance for the Projection (VIP) calculated from OPLS-DA ≥ 1 ; (ii) False Discover Rate (FDR) of paired-t test in WW and DS conditions ≤ 0.05 ; (iii) $|\text{Fold Change (DS/WW)}| \geq 2$. (Reply Figure 1). We added the association analysis of HDMBOA-Glc to the new Figure S8 and discussed it in the revised manuscript lines 328-331 as follows:

“As a product of DIMBOA-Glc, HDMBOA-Glc also showed significant association with *Bx12* loci no matter with or without drought stress (Additional File 1, Figure S8b). Interestingly, HDMBOA-Glc was not a drought-responsive metabolite based on our standard.”

Reply Figure 1. HDMBOA-Glc accumulation in the maize association population (385 inbred lines).

Fig S8 Associations of *Bx11* and *Bx12* loci with DIMBOA-Glc levels.

(a)-(c) Manhattan plot showing the associations of SNPs in region Chr1 66-67 Mb with metabolites HBOA (a), HDMBOA-Glc (b) and DIMBOA-Glc (c). (c) and (d) Associations of SNPs in *Bx11* (c) and *Bx12* (d) loci with metabolite DIMBOA-Glc. (e) and (f) Gene models of *Bx11* (e) and *Bx12* (f). (g) LDs among the significant SNPs of *Bx11* and *Bx12* loci. The significant SNPs in *Bx11* and *Bx12* loci are belonging to the same LD block ($R^2 > 0.2$). On average, SNPs in *Bx12* locus show higher LD to each other than those SNPs in *Bx11* locus.

9. Figure 4a is misleading because it suggests that only HDMBOA-Glc is broken down by a glucosidase. There would be similar glucosidase-mediated breakdown of

DIBOA-Glc, DIMBOA-Glc, DIM2BOA-Glc, and HDM2BOA-Glc.

Reply: Thanks for the reminder. As suggested, we revised Figure 4a with more information of glucosidase-mediated breakdown of TRIBOA-Glc, DIMBOA-Glc, DIM2BOA-Glc, and HDM2BOA-Glc.

Fig 4a. The biosynthetic pathway of benzoxazinoids. Bx12 and other Bx and ZmGLU enzymes are indicated in the pathway.

Table S15 lists about 1,500 genes that are regulated by ZmGLK44, but there is no way to determine which of these genes the authors placed in the different categories that are listed as overexpressed in Figure 5c. Without know this, it is not possible to determine the overall value of the gene categories listed in Figure 5c. Perhaps the genes in Table S15 could be marked to show which category they belong to in Figure 5c.

Reply: Thanks for this suggestion. We added the annotation information of the genes and the enriched pathways for the overexpressed genes in the revised Table S17 (previous Table S15).

10. Table S15: At the bottom of this table there are some genes with an expression level of zero with or without ZmGLK44 expression. How can these be considered differentially expressed?

Reply: Thank you very much for the careful reading and checking on this. We did RNA-

seq for *ZmGLK44* transgenic plants driven by a drought-inducible promoter under well-watered and drought-stressed conditions. Although most genes showing up-regulation were detected in the positive as compared to the negative transgenic plants under drought-stressed conditions, we still observed very few genes differentially expressed under well-watered conditions, which may show no difference under drought-stressed conditions, and these genes were not considered in this study. We are sorry to wrongly list them in the previous Table S15, which caused confusion. We deleted these few genes from the new revised Table S17 (previous Table S15).

11. Figure 5c: Glycerophosphodiester degradation is listed as an enriched category. What genes are actually in this category? Is it possible that this is also tryptophan biosynthesis? A key step in tryptophan biosynthesis is indole-3-glycerolphosphate degradation.

Reply: We listed the genes that were enriched in the cornCyc pathway “glycerophosphodiester degradation” in the revised Table S17. According to the functional annotation, these genes might have no roles in regulation of tryptophan biosynthesis. Please see the revised Table S17.

Table S17. RNA deep sequencing of *ZmGLK44*-OE positive and negative siblings under drought conditions.

^a (+) positive transgenic lines; (-) negative transgenic siblings.

^b The categories were download from cornCyc 9.0 (<https://cornCyc-b73-v3.4.maizegdb.org/>)

Gene	ZmGLK44 -OE#10 (+), DS (TPM) ^a			ZmGLK44 -OE#10 (-), DS (TPM) ^a			Gene Annotation	Categories in Fig 5c ^b
	Repeat1	Repeat2	Repeat3	Repeat1	Repeat2	Repeat3		
GRMZM2G028307	12.86	13.85	26.7	56.6	43.63	8.8	Melibiase family protein	ajugose biosynthesis II (galactinol-independent)
GRMZM2G095126	57.06	55.88	67.98	190.82	173.58	26.99	alpha-galactosidase 1	ajugose biosynthesis II (galactinol-independent)
GRMZM2G063756	9.32	10.68	5.73	0.7	1.34	1.1	Bx5, cytochrome P450, family 71, subfamily A	DIBOA-glucoside biosynthesis
GRMZM2G167549	7.07	7.27	4.56	0.1	0	0.19	Bx3, cytochrome P450, family 71, subfamily A	DIBOA-glucoside biosynthesis
GRMZM2G463996	3.67	3.79	3.53	1.24	1.27	1.9	UDP-Glycosyltransferase superfamily protein	DIBOA-glucoside biosynthesis
GRMZM2G017550	21.14	20.9	21.04	4.83	12.62	24.32	PLC-like phosphodiesterases superfamily protein	glycerophosphodiester degradation
GRMZM2G018820	901.12	828.26	947.77	239.87	448.01	1335.82	senescence-related gene 3	glycerophosphodiester degradation
GRMZM2G034611	20.43	18.34	26.27	7.07	4.55	2.15	suppressor of npr1-1 constitutive 4	glycerophosphodiester degradation
GRMZM2G064750	16.31	15.49	19.49	5.52	2.23	2.84	Protein kinase superfamily protein	glycerophosphodiester degradation
GRMZM2G333045	8.64	7.68	12.36	4.12	1.6	1.64	suppressor of npr1-1 constitutive 4	glycerophosphodiester degradation
GRMZM2G395778	4.38	4.32	5.71	0.28	0.4	0	Protein kinase superfamily protein	glycerophosphodiester degradation
GRMZM2G473511	2.14	1.85	2.51	0.06	0.07	0.12	Protein kinase superfamily protein	glycerophosphodiester degradation
GRMZM2G502350	45.1	54.27	78.26	9.1	3.29	4.08	receptor serine/threonine kinase, putative	glycerophosphodiester degradation
GRMZM5G825193	39.66	50.85	60.09	19.15	10.95	8.1	Protein kinase family protein	glycerophosphodiester degradation

12. Figure 5c: Both tryptophan biosynthesis and benzoxazinoid biosynthesis are differentially regulated by *ZmGLK44*. However, based on other information provided by the authors, this could have opposite effects on drought tolerance. Tryptophan increases drought tolerance and benzoxazinoids (DIMBOA-Glc a downstream product of DIBOA-Glc, a low-abundance intermediate) decreases drought tolerance. It would be good to consider more carefully which benzoxazinoid biosynthesis genes are induced by *ZmGLK44* and not just focus on tryptophan biosynthesis. For instance, is benzoxazinoid biosynthesis up-regulated or down regulated by *ZmGLK44*? What about Bx12 specifically?

Reply: Thanks for this suggestion. We should give more details about the information

of gene expression and benzoxazinoid biosynthesis in the previous manuscript. As suggested, we analyzed the *Bx* gene expression and conducted metabolite profiling of *ZmGLK44* transgenic plants. As shown in revised Figure S12 and Table S17, which was cited in the revised manuscript line 407, the expression of *Bx3* and *Bx5* in the DIBOA-Glc biosynthetic pathway (Richter et al., 2021), and accordingly the contents of DIBOA-Glc were upregulated in positive *ZmGLK44* overexpression plants (+) as compared to their negative siblings (-) under drought conditions. In addition, the other *Bx* genes, including *Bx12*, were not expressed differentially and there is no significant difference of other benzoxazinoids as shown in Figure 4a in positive *ZmGLK44* overexpression plants (+) as compared to their negative siblings (-) under drought conditions.

Fig S12. Upregulation of *Bx* genes and DIBOA-Glc in *ZmGLK44* overexpression plants.

(a) The biosynthetic pathways of Tryptophan, DIBOA-Glc and volatile indole from indole-3-glycerolphosphate as modified based on a previous report (Richter et al., 2021). ER, endoplasmic reticulum. Arrows indicate the upregulation of the genes and metabolite. (b) and (c) The expression of *Bx3* and *Bx4* (b), and the contents of DIBOA-Glc (c) in positive *ZmGLK44* overexpression plants (+) as compared to their negative siblings (-) under drought conditions. Statistical significance was determined by student *t*-test.

Table S17. RNA deep sequencing of *ZmGLK44*-OE positive and negative siblings under drought conditions.

^a (+) positive transgenic lines; (-) negative transgenic siblings.

^b The categories were download from cornCyc 9.0 (<https://cornCyc-b73-v3.4.maizegdb.org/>)

Gene	ZmGLK44 -OE#10 (+), DS (TPM) ^a			ZmGLK44 -OE#10 (-), DS (TPM) ^a			Gene Annotation	Categories in Fig 5c ^b
	Repeat1	Repeat2	Repeat3	Repeat1	Repeat2	Repeat3		
GRMZM2G028307	12.86	13.85	26.7	56.6	43.63	8.8	Melibiase family protein	ajugose biosynthesis II (galactinol-independent)
GRMZM2G095126	57.06	55.88	67.98	190.82	173.58	26.99	alpha-galactosidase 1	ajugose biosynthesis II (galactinol-independent)
GRMZM2G063756	9.32	10.68	5.73	0.7	1.34	1.1	Bx5 cytochrome P450, family 71, subfamily A	DIBOA-glucoside biosynthesis
GRMZM2G167549	7.07	7.27	4.56	0.1	0	0.19	Bx3 cytochrome P450, family 71, subfamily A	DIBOA-glucoside biosynthesis
GRMZM2G463996	3.67	3.79	3.53	1.24	1.27	1.9	UDP-Glycosyltransferase superfamily protein	DIBOA-glucoside biosynthesis

Reference

Richter A, Powell AF, Mirzaei M, Wang LJ, Movahed N, Miller JK, Piñeros MA, Jander G. **Indole-3-glycerolphosphate synthase, a branchpoint for the biosynthesis of tryptophan, indole, and benzoxazinoids in maize.** *Plant J.* 2021;106(1):245-257.

13. Figure 5d: It would be good to also connect this pathway to DIBOA-Glc biosynthesis, which is listed as enriched in Figure 5c. Indole is a branch point for the biosynthesis of tryptophan and benzoxazinoids. See for instance: Richter et al, *Plant Journal*, 2021, 106:245-257. doi: 10.1111/tjp.15163

Reply: Thanks for this suggestion. Yes, we did more analyses to show the gene expression and benzoxazinoids accumulation as suggested. Please see our response to the above question # 12 raised by this reviewer.

14. Line 403-404: "(h) Activation of TRP2p1::LUC, but not the other promoter fragments, by *ZmGLK44* in maize protoplasts." In the figure TRP2p2::LUC is activated (not TRP2p1::LUC).

Reply: Thanks for your careful reading and checking on this. *TRP2*, the tryptophan synthase beta chain 2 encoding gene, was previously named *TSB2* (Richter et al., 2021), we thus changed the name *TRP2* into *TSB2* in our revised manuscript to keep the consistence of this gene. For the same reason, *TRP1*, phosphoribosylanthranilate transferase 1 encoding gene, was renamed *PAT1* in the revised manuscript. We are sorry for the typo and revised it in lines 425-426 as follows:

“(h) Activation of *TSB2p2::LUC*, but not the other promoter fragments, by *ZmGLK44* in maize protoplasts.”

15. Figure 5e: There are at least two tryptophan synthase beta subunit (*TRP2*) genes in maize. Which one is shown in this figure?

Reply: We are sorry for not giving the detailed information of these genes. *ASA2* is GRMZM2G161337, *TRP1* (renamed *PAT1*) is GRMZM2G051219, and *TRP2*

(renamed *TSB2*) is GRMZM2G005024. Please see the revised manuscript in lines 418-419: “(d) Main enzymes (*ASA2*-GRMZM2G161337, *PATI*-GRMZM2G051219 and *TSB2*-GRMZM2G005024) and metabolites involved in Trp biosynthesis process.”

16. Figure 5e,h: There are at least three genes that encode tryptophan synthase alpha subunit activity (indole synthase; TRP1 in the figure) in maize. The authors should specify which gene they are looking at here. Ideally, the authors should look at expression of all three indole synthase genes (TSA, BX1, and IGL).

Reply: The genes we analyzed are *ASA2*-GRMZM2G161337, *PATI*-GRMZM2G051219 and *TSB2*-GRMZM2G005024, which are shown in the revised manuscript lines 417-418. The other three genes did not show significant difference between positive transgenic plants and the negative siblings. However, two *Bx* genes, *Bx3* and *Bx5*, were upregulated in the positive transgenic plants as compared to the negative siblings, and their analyses were shown in revised Fig S12.

17. Figure 5d. TRP1 and TRP2 (tryptophan synthase alpha and beta subunits) are more commonly called TSA and TSB in maize. See for instance: Kriechbaumer et al, BMC Plant Biology, 2008, 8:44. doi: 10.1186/1471-2229-8-44.

Reply: Thanks for the reminder. As suggested, we rename *TRP2* (GRMZM2G005024) *TSB2*, encoding tryptophan synthase beta subunit 2, to make it consistent with the previous publications (Kriechbaumer et al., 2008; Richter et al., 2021). *TRP1* (GRMZM2G051219), phosphoribosylanthranilate transferase 1 encoding gene, was renamed *PATI* in the revised manuscript. Please see our revised manuscript.

Reference

Kriechbaumer V, Weigang L, Fiesselmann A, Letzel T, Frey M, Gierl A, Glawischnig E. **Characterization of the tryptophan synthase alpha subunit in maize.** *BMC Plant Biol.* 2008; 8:44.

Richter A, Powell AF, Mirzaei M, Wang LJ, Movahed N, Miller JK, Piñeros MA, Jander G. **Indole-3-glycerolphosphate synthase, a branchpoint for the biosynthesis of tryptophan, indole, and benzoxazinoids in maize.** *Plant J.* 2021;106(1):245-257.

18. Line 494,499 lepidopteran should not be italicized

Reply: Thanks for the reminder. We revised it as suggested. Please see line 518-519: “Feeding by **lepidopteran** species can induce DIMBOA-Glc methylation to form

HDMBOA-Glc, which enhances the resistance of maize to insect herbivores”

19. Line 498-499: "breeding maize cultivars with both drought tolerance and insect (e.g. lepidopteran species) resistance." This would likely cause sensitivity to other insects (aphids).

Reply: We changed our statement to make it more precision in the revised manuscript lines 522-524 as follows: “*Bx12* and its natural variation CACTA TE could be potential genetic resources or molecular markers in breeding maize cultivars with both drought tolerance and lepidopteran species resistance.”

20. Lines 687-680: "LC-MS data were obtained using a Waters Acquity UPLC 679 system (Waters, <http://www.waters.com>), coupled to a Thermo Fisher Q Extractive (Thermo Fisher, <http://www.thermofisher.com>)." This is not enough information. The authors need to specific what column was used, what gradient, what mass ranges were measured, etc. I should be able to repeat this experiment based on what is written in the methods section.

Reply: Thank you for this suggestion. We updated the method of non-targeted LC-MS/MS. Please see lines 715-737 in our revised manuscript:

“Metabolic Profiling based on Liquid Chromatography-Mass Spectrometry. Metabolites extraction and measurement of maize leaves were performed with LC-MS method essentially as described by Si et al. [66]. 20 mg freeze-dried powder of each sample was mixed with 1 ml pre-cooled methanol/methyl-tert-butyl-ether (1: 3), shaken for 15 mins at 4°C. Then the homogeneity was incubated in an ice-cooled ultrasonication bath and was supplemented with a mixture of 650 ul methanol/water (1: 3). After vortexed and centrifugal for 5 mins at 12,000 rpm in 4°C, the sample was in a phase separation and a 500-ul aliquot from the lower aqueous phase was taken and dried in vacuum. The dried samples were then shipped to Beer-Sheva, Israel for metabolic profiling. After resuspension with 100ul 80% MeOH, the samples were analyzed with a HSS T3 C₁₈ reversed-phase column (100 mm x 2.1mm x 1.8 μm particles; Waters) on a Waters Acquity UPLC system Waters, (<http://www.waters.com>). The mobile phases were 0.1% formic acid in H₂O (A) and 0.1% formic acid in acetonitrile (B). The injection volume is 5μl. The gradient was: 99:1 V/V at 0 min, 99:1 V/V at 1min, 60:40 V/V at 11 min, 30:70 V/V at 13 min, 1:99 V/V at 15 min, 1:99 V/V at 16:00, 99:1 V/V

at 17min, 99:1 at 20 min; flow rate, 0.4 ml/min; temperature: 40°C .

The mass spectra were acquired by a Thermo Fisher Q Extractive (Thermo Fisher, <http://www.thermofisher.com>). Each extract was separately analyzed with both positive and negative modes of electron spray ionization, covering a mass range from 100 to 1500 m/z. The resolution was set to 70,000. The capillary voltage was set to 3.5 kV with a sheath gas flow of 60 and an auxiliary gas flow of 20 (values are in arbitrary units). The capillary temperature was set to 275°C and the drying gas temperature was set to 300°C. In addition to the full MS scan, another MS scan focused on the top-3 features from each previous scan was performed with the normalized collision energy 25.”

21. Lines 686-687: "The further processing of the MS data included isotope clustering, adduct detection, and library search." Again, not nearly enough information. I would not be able to repeat this analysis based on the information that is provided.

Reply: As suggested, we updated the method of pretreatment of chromatography. Please see lines 738-751 in our revised manuscript:

“Chromatograms from the UPLC-LC-ESI-MS/MS runs were analyzed and processed with Refiner MS 11.0.3 (GeneData, <http://www.genedata.com>). Molecular masses, retention times, and associated peak intensities for each sample were extracted from .raw files. Chemical noise was subtracted automatically and a pairwise alignment-based tree was applied in the alignment of chromatogram with the *m/z* windows of five points and retention-time windows of five scans within a RT search interval of 0.5 minute. Peaks originating from the same molecule different isotope patterns were clustering on the basis of isotope patterns using “Chromatogram Isotope Clustering” activity in Refiner MS (RT tolerance: 0.015 min, *m/z* tolerance: 5ppm). Clusters corresponding to the same molecule with different adducts were performed by “Adduct Detection” activity with mass tolerance as 50 ppm and RT tolerance as 0.1 min, the adduct list in negative ionization mode is $[M-H]^-$ and $[M +Hac-H]^-$ while the list is $[M+H]^+$, $[M+Na]^+$ and $[M + NH_4]^+$ in positive ionization mode. Day normalization and sample-median-normalization were conducted and the resulting data matrices were used for further analysis.”

22. Methods section: Although there is a description of the exogenous tryptophan treatment, there is no description of the method that was used for exogenous DIMBOA treatment. Given the reactive nature of DIMBOA and its spontaneous degradation to other metabolites, it should also be noted where DIMBOA was purchased and what purity it had at the time that it was added to maize plants. The half-life of DIMBOA in an aqueous solution has been reported to be about 24 hours.

Reply: Thanks for this suggestion. We updated the method of exogenous DIMBOA treatment. Please see lines 927-938 in our revised manuscript:

“Exogenous DIMBOA treatment was performed with 10-day-old maize B73 seedlings grown under the same condition as plants in L-tryptophan treatment. DIMBOA (Rechemscience, Shanghai) were dissolved in DMSO to 10mg/ml and diluted to 20ug/ml with water. 200ml of DIMBOA solution with 0.1% Tween-20 was sprayed onto maize leaves (n = 72) for five days and the controlled plants were sprayed with solution contained the same amount of DMSO and Tween-20. Drought treatment, survival rate investigation and statistical analysis process of these plants were the same with plants treated with L-tryptophan. For plants grown under well-watered conditions, 40 DIMBOA-treated and control plants were marked and their height were measured at 5 and 10 days after DIMBOA feeding. Top expand leaves of three plants were harvested and mixed with H₂O₂ and MDA measurement as previously described. Shoot of four plants were harvest, weighted (fresh weighted), dried at 65°C for constant weight to obtain dry mass.”

23. Table S3 lists only the 1035 metabolites that the authors deem to be drought-responsive. For the scientific record, negative evidence is also important. Also, some readers may have an interest in a particular mass feature that the authors did not conclude to be drought-responsive. Please include all 3,890 reliably detected mass features in this table.

Reply: Thanks for this suggestion. We revised Table S4 and now it contains all 3,890 mass features detected in our study. The drought-responsive mass features were also indicated in the revised Table S3.

24. Table S5. What about the mQTL for the metabolites that the authors did not deem to be drought-responsive? There is a lot of information that would results from a more complete analysis of the metabolomic data set that is just being ignored. The 1035 chosen metabolites are just based on some arbitrary cutoff that the authors have

determined. Readers will also be interested in the mQTL affecting the other 2,855 metabolites that were detected in these assays. Please include this information in Table S4.

Reply: As suggested, the information of significant SNP markers associated with all 3890 metabolite traits were added to the revised Table S5 (previous Table S4).

25. Table S14. A putative metabolite is described as "34N" (in cell C10 of the spreadsheet). That is not a very informative metabolite description.

Reply: Thanks for this suggestion. This metabolite was annotated based on the database KNApSACk. The information of this metabolite could be reached at http://www.knapsackfamily.com/knapsack_core/information.php?word=C00018512. We search the InChIKey in pubchem and this metabolite is named N-Acetyl-beta-oxotryptamine (<https://pubchem.ncbi.nlm.nih.gov/compound/4188168>). In the revised manuscript, we describe this metabolite as N-Acetyl-beta-oxotryptamine. Please see the revised manuscript line 415 “34N: N-Acetyl-beta-oxotryptamine” and Table S16.

26. Table S14: PP_Group_00384 is described as likely being tryptophan (line 7). However, in line 41 a different metabolites is described as being tryptophan.

Reply: Thank you very much for your careful reading and checking on this, we are sorry for the confusion. The metabolic analyses of *ZmGLK44* transgenic plants were conducted two times with different purposes. The first-time analysis was to check the roles of *ZmGLK44* in regulation of the metabolites as shown in Fig 5a or in Table S16 lines 7-30. For the second-time analysis, we harvested new samples of *ZmGLK44*-overexpression plants for metabolite profiling, and focused on investigating *ZmGLK44*-mediated regulation of Trp and Trp derivatives as shown in Table S16 lines 31-44. Based on these two experiments, we obtained a general trend for Trp accumulation, that is, Trp was more highly enhanced in production in *ZmGLK44*-overexpressing positive plants as compared to their negative non-transgenic siblings after drought stress (Fig 5b, f; Table S16), indicating a positive role of *ZmGLK44* in Trp accumulation under drought conditions. To avoid unnecessary confusion, we removed the Trp data obtained from the second-time experiment.

27. Line 879-880: "All other reasonable requests for data and research materials are available by contacting the corresponding authors." All raw data for large data sets, e.g. RNAseq and metabolomics should be placed in a public database. RNAseq data should

be in GenBank short read archive. Metabolomic raw data (.raw files) should be placed in Dryad, Cyverse or some other public database. If ten or twenty years from now someone wants to look at these data, there is no certainty that the authors will still have the data and will be able to respond to a "reasonable request."

Reply: Thanks for the reminder. We submitted the RNA-seq data to Genome Sequence Archive (GSA), which can be reached at <https://ngdc.cncb.ac.cn/gsa> with the accession number CRA004467, and deposited metabolomic raw data to Omix, which is available at <https://ngdc.cncb.ac.cn/omix> with the accession number OMIX419. Please see line 976-985 in revised manuscript Availability of data and materials section.

28. Is the large SNP dataset for the inbred line population available from a public database? If so, which database? If not, it should be made available.

Reply: The SNP dataset has been published previously (Yang et al., 2014), and we added the download link in line 985: <http://maizego.org/Resources.html>.

Reference

Yang N, Lu Y, Yang X, Huang J, Zhou Y, Ali F, Wen W, Liu J, Li J, Yan J. **Genome wide association studies using a new nonparametric model reveal the genetic architecture of 17 agronomic traits in an enlarged maize association panel.** *PLoS Genet.* 2014;10(9):e1004573.

29. The English-language writing is generally good, but there are occasional lapses. I recommend that one of the native English-speaking co-authors should do some careful proofreading. For instance, a sentence like "Of the upregulated metabolites, there were tryptophan (Trp) and its relative metabolite Achillamide derivatives." is not quite grammatically correct.

Reply: Thanks for this suggestion. The English-speaking co-authors carefully edited this manuscript (edited words / sentences were marked with red color), including the exemplified sentence in lines 403-404: "The upregulated metabolites included tryptophan (Trp) and two tryptophan-related metabolites achillamide derivatives."

Reviewer #2: Genomic basis underlying the metabolome-mediated drought adaptation of maize

The Abstract should include which tissues or organs of the maize plant were profiled for metabolite and transcript abundances.

Reply: Thanks for this suggestion. We added the tissue information for metabolite and transcript profiling in the revised manuscript lines 28-29 as follows:

“we performed non-targeted metabolic profiling of **leaves** for 385 maize natural inbred lines grown under well-watered as well as drought-stressed conditions”, and line 35-38 as follows:

“The regulatory variants that controlled the expression of the candidate genes were revealed by expression QTL (eQTL) analysis to the transcriptomes of **leaves** from 197 maize natural inbred lines in WW and DS”.

Line 110, change from principle to principal component analysis

Reply: Thanks for your careful reading. Sorry for the typo. We revised it as suggested. Please see lines 111-114 in our revised manuscript:

“We conducted **principal** component analysis (PCA) for the maize inbred lines with all these 3,890 metabolites. The results showed a clear separation between the WW and DS conditions, with that **principal** component (PC) 1 explaining the largest proportion (16.37%) of the observed phenotypic variance”.

Lines 136-139, this is a far too sweeping statement especially given that this is an artificial "drought" study in pots. It needs to be couched as upregulated for this environment. Let's see the results across multiple field environments before concluding as such. Also, there is way to conclude that these shifts in metabolites are causal in conferring drought tolerance. They are responding. That is all that can be concluded.

Reply: The big challenge for drought experiments in the field is the control of the weather. Alternatively, we performed the drought experiments in the growth pools to maximally mimic the field conditions, where the growth conditions can be easily controlled. As shown in reply Figure 2 and Figure S1a, the square of each growth pool is 3 m x 7 m = 21 m². Each pool consists of two layers: the top layer is for maize growth, while the bottom layer is for drainage. There are many holes equally distributed in the bottom of the top pool to allow for quick water drainage. A layer of geotextile was placed between the bottom of the top pool and the soil, to prevented soil from falling through the holes. However, we agree with this reviewer that these metabolites are drought responsive. Therefore, we changed our statement as suggested in the revised manuscript lines 138-141 as follows:

“Notably, most (83.3%, or 862/1035) metabolites showed up-regulation patterns in response to drought (Fig. 1c), indicating that up-regulation of genome-wide metabolites

tends to have dominant roles in control of maize drought response, which might confer plant drought tolerance.”

Reply Figure 2. Diagrammatic representation of the growth pools used for maize growth and treatments. The growth pools were made of concrete. Each pool consists of two layers: the top layer is for maize growth, while the bottom layer is for drainage. There are many holes equally distributed in the bottom of the top pool to allow for quick water drainage. A layer of geotextile was placed between the bottom of the top pool and the soil, to prevent soil from falling through the holes.

Fig S1a. Plant cultivation in the growth pools. WW, well-watered; DS, drought-stressed.

Lines 151-154, these rrBLUP results with metabolites would be better served with the inclusion of a comparison to prediction of the dry mass, fresh weight, and survival rate phenotypes with genome-wide genetic markers. This population has been scored with SNP markers, so it is a straightforward comparison. The focus on significance here is misleading, as the focus should be on prediction ability/accuracy.

Reply: Thanks for the suggestion. Drought is a very complex agronomic trait, which is control by many genes with small effects (Hu and Xiong, 2014). The prediction

accuracies (r) by drought-responsive metabolites reached ~ 0.4 for survival rates and even higher for biomass (Fig 1e), indicating that drought-responsive metabolites could efficiently reflect the response and tolerance of maize to drought. As suggested, we performed rrBLUP with SNPs associated with all 3,890 metabolites (s2) and 1,035 drought-responsive metabolites (s1) to predict the survival rates, which have been widely used to monitor maize drought tolerance (Liu et al., 2013; mao et al., 2015; Wang et al., 2016; Xiang et al., 2017). Missing markers were imputed by the function *A.mat()* in R package rrBLUP. The results were shown in Reply figure 3a. Again, the prediction accuracy using SNPs associated with 1,035 drought-responsive metabolites is significantly higher than those using other SNPs. Interestingly, the prediction accuracy with drought-responsive metabolites is higher than that with SNPs associated with 1,035 drought-responsive metabolites (Reply figure 3b), which might because that metabolites play more direct roles than genes in drought response and tolerance. We did focus on the prediction accuracies, but used significance to compare the difference of prediction accuracies with different metabolites. Please see the manuscript in lines 153-155:

“The results showed that the prediction accuracy for drought tolerance, especially for survival rate, using 1,035 drought-responsive metabolites, was significantly higher than those using the total or randomly selected metabolites.”

Reply Figure 3. Comparison of the prediction accuracies of survival rates by metabolites and SNPs associated the metabolites via the rrBLUP approach. a. The prediction accuracies of dry masses, fresh weights and survival rates by all, drought-responsive, and randomly selected metabolites in the maize population after drought

stress via the rrBLUP approach. **b.** The prediction accuracies of dry masses, fresh weights and survival rates by SNPs associated with all, drought-responsive, and randomly selected metabolites (as shown in **a**) in the maize population after drought stress via the rrBLUP approach.

Reference

- Hu H, Xiong L. **Genetic engineering and breeding of drought-resistant crops.** *Annu Rev Plant Biol.* 2014; 65:715-41.
- Liu S, Wang X, Wang H, Xin H, Yang X, Yan J, Li J, Tran LS, Shinozaki K, Yamaguchi-Shinozaki K, Qin F. **Genome-wide analysis of *ZmDREB* genes and their association with natural variation in drought tolerance at seedling stage of *Zea mays L.*** *PLoS Genet.* 2013;9(9):e1003790.
- Mao H, Wang H, Liu S, Li Z, Yang X, Yan J, Li J, Tran LS, Qin F. **A transposable element in a *NAC* gene is associated with drought tolerance in maize seedlings.** *Nat Commun.* 2015;6:8326.
- Wang X, Wang H, Liu S, Ferjani A, Li J, Yan J, Yang X, Qin F. **Genetic variation in *ZmVPP1* contributes to drought tolerance in maize seedlings.** *Nat Genet.* 2016;48(10):1233-41.
- Xiang Y, Sun X, Gao S, Qin F, Dai M. **Deletion of an Endoplasmic Reticulum Stress Response Element in a *ZmPP2C-A* Gene Facilitates Drought Tolerance of Maize Seedlings.** *Mol Plant.* 2017; 10(3):456-469.

How is this statistical test carried out anyway? It should not be a t-test. The rationale for not using a t-test is that the magnitude of the denominator of the t-statistic is dependent on the number of resampling runs (standard error of the difference in means). Therefore, if performing a sufficient number of resampling runs, the denominator would be very small, and it could then be possible to obtain a significant p-value for any comparison with an arbitrarily small difference. Another way of understanding the problem with the t-test is that because each fold is drawn from the same original dataset, the different folds are not independent of each other, which violates the t-test assumption that observations are independent. I suggest using a bootstrap significance testing approach, which I believe to be valid in this case. If properly conducted, it should be independent of the number of resampling runs.

Reply: Thanks for the valuable suggestion. As suggested, we performed the bootstrap significance testing approach using the function *boot.pval()* in R package *boot.pval* to examine the difference of prediction accuracy by using different predictors (Canty and

Ripley, 2020; Thulin, 2021). The origin hypothesis is the ratio of two groups of prediction accuracy is equal to 1. Bootstrap replicates were generated using the function *boot()* in R package *boot* and the bootstrap replicate number is 100. The comparison of *t.test* and *boot.pval* was shown in Reply table 1. The P-values were revised in Fig 1e based on *boot.pval* analyses.

Reply table 1. P-value comparison between *t.test* and bootstrap significance testing.

	Survival rate		Fresh weight		Dry mass	
P-Value	m1 vs m2	m1 vs m3	m1 vs m2	m1 vs m3	m1 vs m2	m1 vs m3
t.test	2.14×10^{-94}	2.84×10^{-76}	3.52×10^{-13}	1.47×10^{-36}	9.7×10^{-7}	6.47×10^{-34}
boot.pval	1×10^{-2}	1×10^{-2}	1×10^{-2}	1×10^{-2}	1×10^{-2}	1×10^{-2}

Reference

Thulin M. **boot.pval: Bootstrap p-Values. R package version 0.1.0.** <https://CRAN.R-project.org/package=boot.pval>. 2021.

Canty A and Ripley B. **boot: Bootstrap R (S-Plus) Functions. R package version 1.3-25.** 2020.

Line 162, kinship not kinships

Reply: Thanks for the reminder. We revised it as suggested. Please see lines 164-167 in our revised manuscript: “we performed GWAS to detect the significant associations between SNP markers and the drought-responsive metabolites (or m-traits) with mixed linear model (MLM) controlling population structure (Q) and **kinship** (K) in TASSAL5 (<https://tassel.bitbucket.io/>)”.

Line 164, need to report what multiple test correction approach was used. If none was used, then either use Bonferroni or FDR. Same for eQTL mapping. The 1/X method is too permissive.

Reply: Thank you very much for your careful reading and suggestions to our manuscript. It is true that stricter threshold (e.g. Bonferroni or FDR correction) could reduce false positive results. However, it could also cause the loss of false negative results, especially when there is linkage among many SNP markers, which results in the non-independence of these SNPs. Taking into consideration of the linkage among the SNP makers of the same maize populations used in our study, a previous GWAS

study of these populations in drought response used a threshold of $P < 1 \times 10^{-5}$ to control type I error ($1/n$, n is the total number of SNPs. Wang et al., 2016). To balance the non-independence of the SNP markers and the Bonferroni correction, we also used the threshold $1/n$ (n = total markers used) in GWAS in this study, which also has been used in previous GWAS and eQTL studies (Wen et al., 2014; Liu et al., 2017; Wu et al., 2021).

Notably, we proved the reliability of the candidate genes from metabolite-trait-based GWAS with multiple assays: 1) there were 35 previously-identified drought-tolerant genes were detected in our candidate gene set (Fig. 2a and Table S8), 2) association analysis with survival rate (SR) of this maize panel showed that there were more significant SNP-SR associations in candidate genes than random simulation (Fig. 2b), 3) we successfully validated the roles of two candidate genes *ZmBx12* and *ZmGLK44* in regulation of metabolite traits and maize drought tolerance via genetic and molecular experiments (Fig. 4-6).

References

- (1) Fu J, Cheng Y, Linghu J, Yang X, Kang L, Zhang Z, Zhang J, He C, Du X, Peng Z: **RNA sequencing reveals the complex regulatory network in the maize kernel.** *Nature Commun.* 2013, **4**:2832.
- (2) Liu H, Luo X, Niu L, Xiao Y, Chen L, Liu J, Wang X, Jin M, Li W, Zhang Q: **Distant eQTLs and Non-coding Sequences Play Critical Roles in Regulating Gene Expression and Quantitative Trait Variation in Maize.** *Mol Plant* 2017, **10**:414-426.
- (3) Wang X, Wang H, Liu S, Ferjani A, Li J, Yan J, Yang X, Qin F: **Genetic variation in *ZmVPP1* contributes to drought tolerance in maize seedlings.** *Nature Genetics* 2016, **48**:1233-1241.
- (4) Wen W, Li D, Li X, Gao Y, Li W, Li H, Liu J, Liu H, Chen W, Luo J, Yan J: **Metabolome-based genome-wide association study of maize kernel leads to novel biochemical insights.** *Nature Commun.* 2014, **5**:3438.
- (5) Wu X, Feng H, Wu D, Yan S, Zhang P, Wang W, Zhang J, Ye J, Dai G, Fan Y, Li W, Song B, Geng Z, Yang W, Chen G, Qin F, Terzaghi W, Stitzer M, Li L, Xiong

L, Yan J, Buckler E, Yang W, Dai M. **Using high-throughput multiple optical phenotyping to decipher the genetic architecture of maize drought tolerance.** *Genome Biol.* 2021; 22(1):185.

Lines 175-176, concluding many new putative drought-responsive QTL is a bit of a stretch. By no means was the maize drought linkage/GWAS mapping literature deeply surveyed.

Reply: Thanks for the reminder. We rephrased our statement as suggested. Please see lines 179-180 in our revised manuscript: “Therefore, our m-trait-based GWAS could detect many new putative **drought-related** QTLs in addition to these of known”.

Line 178, need to define "lead SNP". I assume this is a peak SNP - a SNP with smallest P-value at a locus, but it still needs to be defined.

Reply: Thanks for the suggestion, we mean the peak SNP here exactly. We change the description in the revised manuscript. Please see line 182-184: “Based on the **peak** SNPs (SNP with smallest P-value at a locus) and their linkage disequilibrium (LD), a total of 2,589 candidate genes were identified based on the 7,811 significant SNPs.”

Lines 179-180, a statistical test of enrichment needs to be conducted to say anything meaningful about finding 190 TFs.

Reply: Thanks for the valuable suggestion. We applied a Fisher’s exact test to examine whether the TFs in our candidate gene set were enriched as compared to the genome-wide TFs. There were 2,289 TF genes in all 39,625 genes (ratio = 0.058), while the TF gene number in our 2,589 candidate genes was 190 (ratio = 0.073). The *P* value of the Fisher’s exact test is 6.6×10^{-4} , indicating an enrichment of TFs in our candidate gene set. The transcription factor list of maize was accessible at <http://plantfdb.gao-lab.org/>. The transcription factor annotated and total gene number are based on maize genome v3.31. Please see lines 184-187 in our revised manuscript:

“190 genes among them were TFs (Additional File 2, Table S5), which showed a significantly enrichment as compared to the total genome-wide TFs ($p = 6.6 \times 10^{-4}$, Fisher’s exact test). These results suggested the importance of transcriptional variations in maize drought tolerance regulation.”

Fig 2a - nadir view DS Manhattan plot totally unnecessary and less than helpful. Flip

upright.

Reply: Thanks for the reminder. As suggested, we flipped the DS Manhattan plot upright. Please see revised Fig 2a.

Fig 2(a) Collection of the GWAS signals for the 1,305 drought-responsive metabolites detected under WW (up panel) and DS (bottom panel) conditions. The dashed lines indicate the threshold of significance. The loci without significant SNPs were filtered out. All 35 candidate genes with known drought tolerance roles were annotated based on the significant GWAS signals and indicated in the panels.

Line 192, I have never heard of it referred to as a permutation assay. Permutation test?

Reply: We revised it as suggested. Please see lines 200-201 in our revised manuscript: “The significance was calculated based on 1000 times of permutation test.”

Lines 212-217, I do not agree with the candidate gene only analysis in this context. It is more important to see how these associated genes rank genome-wide rather than compared to a random subset. I am not convinced on reliability of results as stated in lines 217-219 based on the collectively results. It is easy to subsample data in different ways with biological hypotheses, but the presentation and argument thereof needs to be better conveyed and supported beyond correlative machinery. Otherwise, it is a weakened argument.

Reply: Thank you for the insightful comments and valuable suggestions. It is a great idea to analyze the rank of our genes genome-wide. We revised the manuscript in lines 224-230 as follows:

“Next, we ranked the genes based on the p-value of the peak SNP of their locus. In

the top 100 genes with most significant SNPs, 19 of them were found in our candidate gene set, while the expectation gene number is 8 (Fisher's exact test, $p = 6.4 \times 10^{-4}$). The number of candidate gene in top 500 gene list is 72, while the expectation gene number is 41 (Fisher's exact test, $p = 3.25 \times 10^{-6}$), and we observed 126 genes in top 1000 gene list, while the expected genes are 82 (Fisher's exact test, $p = 1.35 \times 10^{-6}$). These results indicated that our candidate genes might be enriched in the putative drought-tolerant genes.”

Line 258, 1Mb window really needs to be used for cis-eQTL and is more often the standard than 20kb despite the citation. This is resulting in inflated number of trans-eQTL.

Reply: Different from those selfed or semi-selfed plant species, such as Arabidopsis, rice and cottons etc, maize is an outcross plants and shows more rapid LD decay. The average LD in maize diverse inbred lines was estimated to be 1.5kb (Remington et al., 2001). The overall LD decay in the population used in this study is 500bp ($r^2=0.1$), reaching single gene resolution (Fu et al., 2013; Li et al., 2013). Because of the rapid LD decay, the cutoff for *cis-* / *trans*-eQTL definition in other studies with this maize population has been set as 20kb (Fu et al., 2013; Liu et al., 2020; Pang et al., 2019).

However, we also tried the 1-Mb window size as cutoff to identified *cis-* / *trans*-eQTLs. Although the exact amounts of eQTLs were different, the general conclusion was not changed based on these eQTLs obtained from different cutoffs, that is, *cis* eQTL explained larger phenotypic variations of the expression traits as compared to the *trans* eQTLs (Reply Table 2).

Reply Table 2. Comparison of cis- and trans- eQTL number and effect using cutoff of 20kb and 1Mb

Total	cis-eQTL num	trans-eQTL	cis-eQTL effect	trans-eQTL effect
20kb	1328	12218	27.77%	23.07%
1Mb	4642	8904	25.43%	22.54%
WW	cis-eQTL num	trans-eQTL	effect cis-eQTL	effect trans-eQTL
20kb	590	6568	27.65%	22.87%
1Mb	2219	4939	25.29%	22.40%
DS	cis-eQTL num	trans-eQTL	effect cis-eQTL	effect trans-eQTL
20kb	738	5650	27.87%	23.29%

Reference

Remington DL, Thornsberry JM, Matsuoka Y, Wilson LM, Whitt SR, et al. **Structure of linkage disequilibrium and phenotypic associations in the maize genome.** *Proc. Natl. Acad. Sci. USA.* 2001, 98: 11479–84

Li H, Peng Z, Yang X, Wang W, Fu J, Wang J, Han Y, Chai Y, Guo T, Yang N, et al: **Genome-wide association study dissects the genetic architecture of oil biosynthesis in maize kernels.** *Nat Genet* 2013, **45**:43-50.

Fu J, Cheng Y, Linghu J, Yang X, Kang L, Zhang Z, Zhang J, He C, Du X, Peng Z, et al: **RNA sequencing reveals the complex regulatory network in the maize kernel.** *Nat Commun* 2013, **4**:2832.

Liu S, Li C, Wang H, Wang S, Yang S, Liu X, Yan J, Li B, Beatty M, Zastrow-Hayes G, et al: **Mapping regulatory variants controlling gene expression in drought response and tolerance in maize.** *Genome Biol* 2020, **21**:163.

Pang J, Fu J, Zong N, Wang J, Song D, Zhang X, He C, Fang T, Zhang H, Fan Y, et al: **Kernel size-related genes revealed by an integrated eQTL analysis during early maize kernel development.** *Plant J* 2019, **98**:19-32.

Lines 272-273, static is misleading, as it implies that it is constitutive. Shared? Anyway, something else is needed.

Reply: Thanks for this suggestion. As suggested, we replaced “static eQTL” with “shared eQTL” in our revised Fig 3c and manuscript lines 288-290.

Fig 3(c). The dynamic and shared eQTLs in total, *trans*- and *cis*-eQTL groups.

Lines 285-286, "expression regulation of themselves..." not clear. Please revise for clarity.

Reply: Sorry for the confusion. As suggested, we reworded the sentence in our revised manuscript in lines 300-301: “These data suggested strong effects of *cis* eQTLs on the expression regulation of the genes nearby.”

Second round of review

Reviewer 1

My concerns from the previous review have been addressed in an adequate manner. This manuscript is a significant contribution to research on drought tolerance in maize.

One interesting thing that I noticed in my re-reading of the manuscript:

The authors identify ZmCS1 (citrate synthase 1, GRMZM2G063909) as a drought-related gene. It should be noted that a previous study (Liang et al, *New Phytologist* (2021) 230: 2355–2370; which shares an author with the current manuscript) identified GRMZM2G063909 as a gene conferring salt tolerance (though they called it ZmCS3). Given the frequent overlap in plant drought and salt stress responses, there is probably some connection here.

Maize gene nomenclature issues come up again here (ZmCS1 vs. ZmCS3 in the current manuscript and Liang et al 2021), highlighting the importance that the authors should always provide unambiguous identifiers like the GRMZM gene numbers, in addition to the abbreviated gene names.

Line 721: Convert 12,00 rpm to g-force or provide the radius of the rotor that was used.

Line 730: Should this be “Q Exactive”?

Point-by-point responses to reviewers

Editor's comments

We have made some minor changes to the abstract of the manuscript, to make it better adhere to Genome Biology's house style. Please find the edited version pasted at the foot of this email. Please ensure that the manuscript contains the following sections: Funding and Ethical Approval sections. If ethical approval was not needed for the study, please state in the manuscript that these are not applicable. Finally, please make sure that the following formatting concerns are addressed prior to resubmission:

- Please remove line numbers
- Please ensure that figures are cited in order throughout the manuscript
- Please remove figures and supplementary figures from the main manuscript.

[Additional file]

- We recommend that you compile all supplementary figures in one file, for the convenience of our readers. Please include supplementary figure legends in this file.
- Name supplementary files as 'Additional file X', and cite explicitly by additional file name in the manuscript e.g. 'Additional file 1: Fig. S1'. Please ensure that if you have more than one additional file that they are cited in ascending order within the main body of text.
- Please provide a subsection after the declarations section listing all the additional files including file names (e.g. Additional file 1), titles and a short description of data..

Reply: We really appreciate the editor's valuable suggestions. We revised the MS as suggested, including:

- 1) We revised the abstract as suggested, the Funding and Ethical Approval sections were added into the MS and we removed the line numbers.
- 2) We carefully checked the citation of the figures and tables in the MS.
- 3) We removed figures and supplementary figures from the main manuscript.
- 4) We compiled all supplementary figures in one file.
- 5) We named supplementary files as 'Additional file X' as suggested, and provided a subsection listing all the additional files

Reviewer reports:

Reviewer #1: My concerns from the previous review have been addressed in an adequate manner. This manuscript is a significant contribution to research on drought

tolerance in maize.

Reply: We really appreciate your careful reading, evaluation, and positive comments on our manuscript.

One interesting thing that I noticed in my re-reading of the manuscript:

The authors identify ZmCS1 (citrate synthase 1, GRMZM2G063909) as a drought-related gene. It should be noted that a previous study (Liang et al, New Phytologist (2021) 230: 2355–2370; which shares an author with the current manuscript) identified GRMZM2G063909 as a gene conferring salt tolerance (though they called it ZmCS3). Given the frequent overlap in plant drought and salt stress responses, there is probably some connection here.

Reply: We agree with you that this CS gene (GRMZM2G063909) may have important roles in responding to different abiotic stresses. In future, more experiments are required to deeply investigate how this gene connects to different abiotic stress responses.

Maize gene nomenclature issues come up again here (ZmCS1 vs. ZmCS3 in the current manuscript and Liang et al 2021), highlighting the importance that the authors should always provide unambiguous identifiers like the GRMZM gene numbers, in addition to the abbreviated gene names.

Reply: We added the unambiguous identifier (GRMZM2G063909) for ZmCS1 in the manuscript.

Line 721: Convert 12,00 rpm to g-force or provide the radius of the rotor that was used.

Reply: We convert 12,00 rpm to 13,523 rcf as suggested.

Line 730: Should this be “Q Exactive”?

Reply: We really appreciate your careful reading. Sorry for the typo. Yes, it is “Q Exactive”. We revised it in the manuscript.